# Nutrient Contents and Productivity of *Triticum aestivum* Plants Grown in Clay Loam Soil Depending on Humic Substances and Varieties and Their Interactions

Ahmed A. M. Awad [1],*, Ashraf B. A. El-Taib [2], Atef A. A. Sweed [1] and Aya A. M. Omran [1]

[1] Soil and Natural Resources Department, Faculty of Agriculture and Natural Resources, Aswan University, Aswan 81528, Egypt; atefsweed@agr.aswu.edu.eg (A.A.A.S.); ayaabdelsalam0094@gmail.com (A.A.M.O.)
[2] Agronomy Department, Faculty of Agriculture and Natural Resources, Aswan University, Aswan 81528, Egypt; ashraf.el_taib@agr.aswu.edu.eg
* Correspondence: ahmed.abdelaziz@agr.aswu.edu.eg; Tel.: +20-973-480-245

**Abstract:** Due to an extreme increase in population growth, Egypt suffers from a widening gap in the quantity of imported wheat compared with production and local consumption. Two field trials were conducted during the 2018/2019 and 2019/2020 seasons with three levels of humic substances (HSs) as a foliar spray (1.0, 2.0 and 4.0 g L$^{-1}$; HS$_1$, HS$_2$ and HS$_3$) and three levels (5.04, 7.56 and 10.08 kg ha$^{-1}$; HS$_4$, HS$_5$ and HS$_6$) as a soil application. These were applied three times (30, 45 and 60 days after sowing) in comparison with the control (HS$_0$) to evaluate the performance of three wheat varieties (Seds1 (V$_1$), Misr2 (V$_2$) and Giza168 (V$_3$)) grown in clay loam soil. The experiment was set up according to the split-plot structure in a randomized complete block design; however, the varieties were set as the main plot and treatments were a sub-main plot. Generally, the data indicated that the soil application treatments recorded maximum values for most growth and yield attributes, except for spike length and grain weight per spike, SPAD reading and total grain yield in the first season, and leaf area and biological yield in the second season. HS$_1$, HS$_2$, HS$_5$, and HS$_6$ were the superior treatments for most of the nutrient contents studied. Regarding the influence of variety, the results showed that V$_3$ recorded maximum values for LA, SpL, TGW, TGY and leaf Zn and Cu contents in both seasons; PH, GWS and leaf N content in the first season; and SPAD reading, BY and leaf K, Fe and Mn contents in the second season. V$_1$ was the superior variety for GWS, BY, leaf K and Mn contents in the 2018/2019 season and PH, GNS in the second season, followed by V$_2$, which had the greatest values for leaf P contents in both seasons, and SPAD reading, GNS and leaf Fe content in the 2018/2019 season and GWS and leaf N content in the second season.

**Keywords:** humic substances; wheat varieties; nutrient content; yield and its components

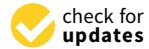



## 1. Introduction

According to the Food and Agriculture Organization of the United Nations [1], wheat (*Triticum aestivum* L.) is one of the most important cereal crops grown all over the world. It supplies nearly 20% of the food calories for the world's population. The total cultivated area of wheat reached approximately $1.425 \times 10^6$ hectare (ha) and a total production about $9.279 \times 10^6$ tons with an average of 6.511 tons per hectare. Because Egypt suffers from a huge shortage in its production of wheat as a result of the extreme increase in population growth, it imports more than 50% of its total wheat consumption. According to the economic affairs sector of Egypt, the total cultivated area reached about 1,306,228 ha and total production reached 8,558,807 tons, with an average of 6.552 tons per hectare. For this reason, we need to increase wheat production to overcome this lack in production by maximizing productivity per unit. Thus, increasing wheat production is considered the main method to reduce the wide gap between (i) the increase in population and (ii) production and consumption, which amounts to 25% of total production.

In recent times, due to increased popular awareness of food quality, citizens have started to search for better food that is free from pollutants. Many developments have been made in terms of improving the quality and quantity of agricultural products, for example, the substitution of some organic fertilizers such as humic substances (HS), whether partially or completely, for chemical fertilizers for increasing nutritional values and decreasing negative environmental effects [2]. Other developments include improving the soil's physical, chemical, and biological properties, especially in Egyptian soils, which suffer from a reduction in soil organic matter (SOM) [3], as well as plant nutrition, crop management and other conditions [4–6].

One of the applications of organic agriculture is the use of organic fertilizer, which can take many forms, such as humic substances (HSs) including humic acid (HA) and fulvic acid (FA) [7,8]. HSs are applied to plants as either a foliar spray or a soil application [9]. HA is the final breakdown component of the natural decay of animals, plants and microorganisms [8]. It plays a vital role in promoting plant growth by boosting metabolic activity. Similarly, it increases nutrient absorption and cell division, which have a direct influence on the productivity of many crops [10–12]. Moreover, HA has a stimulative effect through an interference mechanism on plant metabolism, resulting in influences on the soil and plants. HS is an intimate part and vital constituent of SOM. Many scientists use HS as a soil application and/or a foliar spray to improve soil conditions and plant growth, and to explore the key roles of HA in ameliorating the negative properties of soils in order to enhance plant growth and nutrient uptake. Several studies evaluated the best application method of HA and the optimum dose, either as a foliar spray or as a soil application. HA is known to be among the most biochemically active materials found in soil and is considered to be the most abundant naturally [13].

At present, HSs are widely applied to change nutrient availability in the soil for roots through improving the physio-chemical properties. HA is also a principal component of HS, which is the major organic constituent of soil (humus). Moreover, HSs have several beneficial impacts on soil structure and increase soil microbial populations, as well as modifying the mechanisms involved in plant growth stimulation, plant cell permeability and nutrient absorption, thus enhancing productivity [14,15].

Currently, growers' mentality has changed regarding the application of organic matter, whether of animal or plant origin, as a manure source instead of chemical fertilizers for the purpose of decreasing damage to agricultural soils, the environment and agricultural production due to excessive use of chemical fertilizers and pesticides. HA has several environmental benefits, providing profitable and effective solutions to environmental problems and helping preserve the environment [16]. Furthermore, HA has the advantages of increasing yield and quality. Healthy and environmentally friendly food production is a priority for scientists and researchers. Organic agriculture presents itself as an efficient alternative, although there are concerns about yield.

The ultimate goal of the present study was to investigate the influence of HS on yield and its attributes, as well as the chemical constituents of wheat plants. In addition, the study compared HS applied as a foliar spray and HS applied as a soil application to evaluate the ideal application method under clay loam soil conditions in Aswan Governorate, Egypt.

## 2. Materials and Methods

### 2.1. Location of the Field Experiments, Weather Conditions, Timing and Plant Materials

Two field trials were conducted at Al-Raghama Gharb village, Koum-Ombo district, (24°52′26″ N, 32°56′30″ E), Aswan Governorate, Egypt, in the winter seasons of 2018–2019 and 2019–2020 on three wheat (*Triticum aestivum* L.) varieties with six levels of humic substances (HS). The varieties were produced by the Agricultural Research Center (ARC) in Giza, Egypt. The varieties of wheat tested were Seds1 ($V_1$), Misr2 ($V_2$) and Giza168 ($V_3$), which were sown on 12 December 2018 and 10 December 2019. However, $V_1$ is characterized by high yield in addition to its high tolerance to yellow rust disease. Furthermore, it has low water requirements, and it is also recommended to cultivate it in soils which suffer

from salinity. $V_2$ is characterized by its high production and tolerance to salinity and water deficit, whereas $V_3$ is one of the high-yield varieties that withstands high temperatures and irrigation water deficit compared to other varieties. For these reasons, the selected wheat varieties are the most prevalent in the study region as a result they are more adaptable to the climatic conditions in Upper Egypt. Weather conditions were presented in Table 1.

**Table 1.** The weather data from the Aswan region in Egypt during wheat growing seasons.

| Month | Day °C | Night °C | ARH (%) | AWS (ms$^{-1}$) | AP (mm d$^{-1}$) |
|---|---|---|---|---|---|
| **2018/2019 season** | | | | | |
| December | 27.38 | 4.63 | 47.31 | 2.52 | 0.00 |
| January | 31.84 | 2.08 | 33.38 | 2.19 | 0.16 |
| February | 36.10 | 5.57 | 38.25 | 2.80 | 0.07 |
| March | 36.59 | 5.56 | 26.94 | 2.90 | 0.00 |
| April | 41.55 | 8.87 | 19.69 | 2.92 | 0.00 |
| **2019/2020 season** | | | | | |
| December | 28.78 | 5.53 | 42.94 | 2.38 | 0.00 |
| January | 30.53 | 3.10 | 47.75 | 2.66 | 0.01 |
| February | 31.88 | 4.26 | 39.25 | 2.67 | 0.03 |
| March | 38.94 | 7.02 | 27.12 | 2.88 | 0.00 |
| April | 42.13 | 12.64 | 21.25 | 3.27 | 0.00 |

Day °C = Average day temperature, Night °C = Average night temperature, ARH = Average relative humidity, AWS = Average wind speed and AP = Average precipitation.

### 2.2. Treatment, Experimental Design and Intercultural Operations

The soil texture tested was clay loam soil in both seasons. Three varieties of wheat, i.e., $V_1$, $V_2$ and $V_3$, were tested. Six levels of HS (1.0, 2.0 or 4.0 g L$^{-1}$ as a foliar spray; 5.04, 7.56 and 10.08 kg ha$^{-1}$ as a soil application) were compared with the control treatment, as presented in Table 2. All treatments were applied three times: 30, 45 and 60 days after sowing (DAS). The chemical composition of humic substances applied in the study as shown in Table 3.

The main experimental plots were arranged according to the three wheat varieties. Each main plot was subsequently subdivided into seven treatments of HS. Thus, there were $3 \times 7 = 21$ treatment interactions. The area of each plot was 3 m (width) $\times$ 3.5 m (length) = 10.5 m$^2$. There were 15 rows in each plot and 17–18 hills for sowing seeds in each row. Five seeds were sown in each hill with a distance of 20 cm between hills. However, the sowing rate was 96 kg ha$^{-1}$. The plants were fertilized with N, P and K fertilizers which were used in the form of ammonium sulfate $(NH_4)_2SO_4$ N $\approx$ 20.6%N, calcium superphosphate $(CaH_6O_9P_2)$ $P_2O_5$ $\approx$ 15.5% and potassium sulfate $(K_2SO_4)$ $K_2O$ $\approx$ 48–50% at levels of 180, 36 and 24 kg ha$^{-1}$, respectively, according to the recommendations of the Egyptian Ministry of Agriculture. In both seasons, the field experiment was established according to the split-plot structure for a randomized complete block design (RCBD) with three replicates.

### 2.3. Determination of the Soil Properties

Surface soil samples were collected at a depth of 0–25 cm before sowing and transported to the Soil, Water and Plant Analysis Laboratory (SWPAL) at the Faculty of Agriculture and Natural Resources, Aswan University, to determine some soil physical and chemical properties such as particle size distribution using the hydrometer method [17]. Soil acidity (pH) in soil paste and electrical conductivity (ECe) in a soil paste extract were measured using a pH meter [18]. Electrical conductivity (EC) and calcium carbonate (CaCO$_3$%) were determined using an EC meter and a calcimeter, respectively, as described by [19]. Soil

organic matter (SOM) was determined as described by [20]. Exchangeable cations (sodium ($Ca^{++}$), magnesium ($Mg^{++}$), sodium ($Na^+$) and potassium ($K^+$)) were extracted with 1 N $NH_4AC$; however, $Ca^{++}$ and $Mg^{++}$ were determined using the EDTA titration method, while $Na^+$ and $K^+$ were determined using flame photometry [21]. Exchangeable anions (bicarbonate ($HCO_3^-$), carbonate ($CO_3^-$), chloride ($Cl^-$) and sulfate ($SO_4^-$) were determined with the titration method [18]. Nitrogen, phosphorus and potassium were determined according to the method described by [19]. Available iron (Fe), manganese (Mn), zinc (Zn) and copper (Cu) were extracted with DTPA [22] and determined using inductively coupled plasma–optical emission spectrometry (ICP-OES, PerkinElmer OPTIMA-2001 DV, Norwalk, CT, USA) according to the method described in [23]. The results of the analysis are shown in Table 4.

**Table 2.** The description of the experimental treatments.

| Varieties | Symbol | Dose of HS (Humic Substance) | Symbol | Application Time |
|---|---|---|---|---|
| Seds1 | $V_1$ | No HS applied (spraying with distilled water) | $HS_0$ | |
| | | 1.0 g $L^{-1}$ of HS as a foliar spray | $HS_1$ | |
| | | 2.0 g $L^{-1}$ of HS as a foliar spray | $HS_2$ | |
| | | 4.0 g $L^{-1}$ of HS as a foliar spray | $HS_3$ | |
| | | 5.04 kg $ha^{-1}$ of HS as a soil application | $HS_4$ | |
| | | 7.56 $kg^{-1}$ of HS as a soil application | $HS_5$ | |
| | | 10.08 kg $ha^{-1}$ of HS as a soil application | $HS_6$ | |
| Misr2 | $V_2$ | No HS applied (spraying with distilled water) | $HS_0$ | All treatments were performed three times after 30, 45, 60 days from sowing |
| | | 1.0 g $L^{-1}$ of HS as a foliar spray | $HS_1$ | |
| | | 2.0 g $L^{-1}$ of HS as a foliar spray | $HS_2$ | |
| | | 4.0 g $L^{-1}$ of HS as a foliar spray | $HS_3$ | |
| | | 5.04 kg $ha^{-1}$ of HS as a soil application | $HS_4$ | |
| | | 7.56 $kg^{-1}$ of HS as a soil application | $HS_5$ | |
| | | 10.08 kg $ha^{-1}$ of HS as a soil application | $HS_6$ | |
| Giza168 | $V_3$ | No HS applied (spraying with distilled water) | $HS_0$ | |
| | | 1.0 g $L^{-1}$ of HS as a foliar spray | $HS_1$ | |
| | | 2.0 g $L^{-1}$ of HS as a foliar spray | $HS_2$ | |
| | | 4.0 g $L^{-1}$ of HS as a foliar spray | $HS_3$ | |
| | | 5.04 kg $ha^{-1}$ of HS as a soil application | $HS_4$ | |
| | | 7.56 $kg^{-1}$ of HS as a soil application | $HS_5$ | |
| | | 10.08 kg $ha^{-1}$ of HS as a soil application | $HS_6$ | |

**Table 3.** The chemical composition of applied humic substances in the study.

| Chemical Composition | % | Chemical Composition | % |
|---|---|---|---|
| Humic acid (HA) | 52.5 | Fulvic acid (FA) | 12.5 |
| Nitrogen (N) | 0.95 | Phosphorus (P) | 1.04 |
| Potassium (K) | 1.46 | Calcium (Ca) | 2.81 |
| Magnesium (Mg) | 0.92 | Sulfur (S) | 0.48 |
| Iron (Fe) | 0.61 | Manganese (Mn) | 0.09 |
| Zinc (Zn) | 0.32 | Copper (Cu) | 0.55 |
| Sodium (Na) | 0.04 | Others | 0.44 |

*2.4. Evaluation of Some Growth and Physiological Parameters*

At 90 days after sowing (DAS), the 10 most homogeneous plants were selected from each plot and average plant height was measured (in centimeters) using a graduated rule. For rapid and accurate measurements of the fourth and fifth leaf contents, relative SPAD meter values were determined using a handheld SPAD-502 m device. The leaf area (LA) was measured by a planimeter device.

**Table 4.** Some soil physical and chemical properties of the experimental fields in 2018/2019 and 2019/2020 seasons.

| Soil Property | 2018/2019 | 2019/2020 |
|---|---|---|
| Particle size distribution | | |
| Sand % | 22.0 | 22.2 |
| Silt % | 38.8 | 37.7 |
| Clay % | 39.2 | 40.1 |
| Soil texture class | Clay loam | Clay loam |
| pH in soil paste | 7.66 | 7.96 |
| ECe (dSm$^{-1}$) in soil paste extracted | 0.34 | 0.31 |
| Organic matter (OM) % | 5.99 | 3.99 |
| $CaCO_3$ | 1.07 | 1.09 |
| Soluble ions meq/L soil | | |
| $CO_3^-$ | - | - |
| $HCO_3^-$ | 1.00 | 2.00 |
| $Cl^-$ | 2.50 | 2.50 |
| $Ca^{++}$ | 3.50 | 3.50 |
| $Mg^{++}$ | 1.50 | 1.00 |
| $Na^+$ | 2.99 | 2.80 |
| $K^+$ | 0.54 | 0.37 |
| $SO_4^-$ | 3.94 | 3.17 |
| Total elements | | |
| N % | 0.015 | 0.013 |
| Available-P mg kg$^{-1}$ (Extractable with NaHCO$_3$ pH = 8.5) | 3728.0 | 3815.0 |
| Available K mg kg$^{-1}$ (Extractable with Ammonium acetate pH = 7.0) | 352 | 376 |
| DTPA available Micronutrients | | |
| Fe, mg kg$^{-1}$ | 13.6 | 14.3 |
| Mn, mg kg$^{-1}$ | 1.86 | 1.95 |
| Zn, mg kg$^{-1}$ | 0.11 | 0.15 |
| Cu, mg kg$^{-1}$ | 0.07 | 0.09 |

*2.5. Evaluation of Leaf Nutrients*

At 90 DAS, a random sample of leaves from 10 plants was collected from each plot in the three replicates and washed with distilled water. The leaf samples were oven-dried at 70 °C and crushed to determine the total N, P and K contents according to the method described in [24]. Fe, Mn, Zn and Cu were determined using inductively coupled plasma–optical emission spectrometry (ICP-OES, PerkinElmer OPTIMA-2001 DV, Norwalk, CT, USA) according to the method described in [23].

*2.6. Yield and Its Components*

At harvest, spike length (SpL, cm), grain weight per spike (GWS, g), grain number per spike (GNS, n spike$^{-1}$), 1000-grain weight (TGW, g), biological yield (BY, kg plant$^{-1}$) and total grain yield (TGW, ton ha$^{-1}$) were measured and calculated from 30 random plants in each plot and then calculated as a percentage of the total number of plants per hectare.

*2.7. Statistical Analysis of Variance*

Analysis of variance (ANOVA) and Duncan's test were calculated using the GENSTAT statistical package, version 9.2 (GENSTAT, 2007). Correlation coefficients and stepwise regression were calculated using the IBM SPSS Statistics 21 Wizard.

## 3. Results

*3.1. Influence of Humic Substances, Wheat Varieties and Their Interactions on Plant Height, Leaf Area and SPAD Readings*

### 3.1.1. Influence of Humic Substances

The results (Figures 1–3) indicated that all HS treatments, either as a foliar spray or a soil application, significantly affected wheat plant height (PH) and leaf area (LA). The maximum values for plant height in both seasons (112.49 and 99.08 cm) and for leaf area in the second season (45.40 cm$^2$) were obtained only from plants fertilized with 10.08 kg ha$^{-1}$ of HS as a soil application (HS$_6$), while the maximum value for LA in the first season (46.51 cm$^2$) was produced by the HS$_6$ treatment. Dissimilar data were obtained for the SPAD readings; however, plants nourished with 1.0 (HS$_1$) and 4.0 (HS$_3$) g L$^{-1}$ of HS had the highest values (47.26 and 43.83) in the 2018/2019 and 2019/2020 seasons, respectively. On the other hand, for AS, as shown in Figures 1–3, the data indicated percentages of increase between the highest and lowest values of 17.01 and 3.06 for PH, 9.31 and 5.88 for LA, and 10.63 and 3.57% for SPAD reading in 2018/2019 and 2019/2020, respectively. The results of the ANOVA indicated that all treatments had significant effects (at $p \leq 0.01$) on the PH in both growth seasons and on the SPAD reading in the first season only, but had no significant influence on the LA in both seasons and the SPAD reading in the second season.

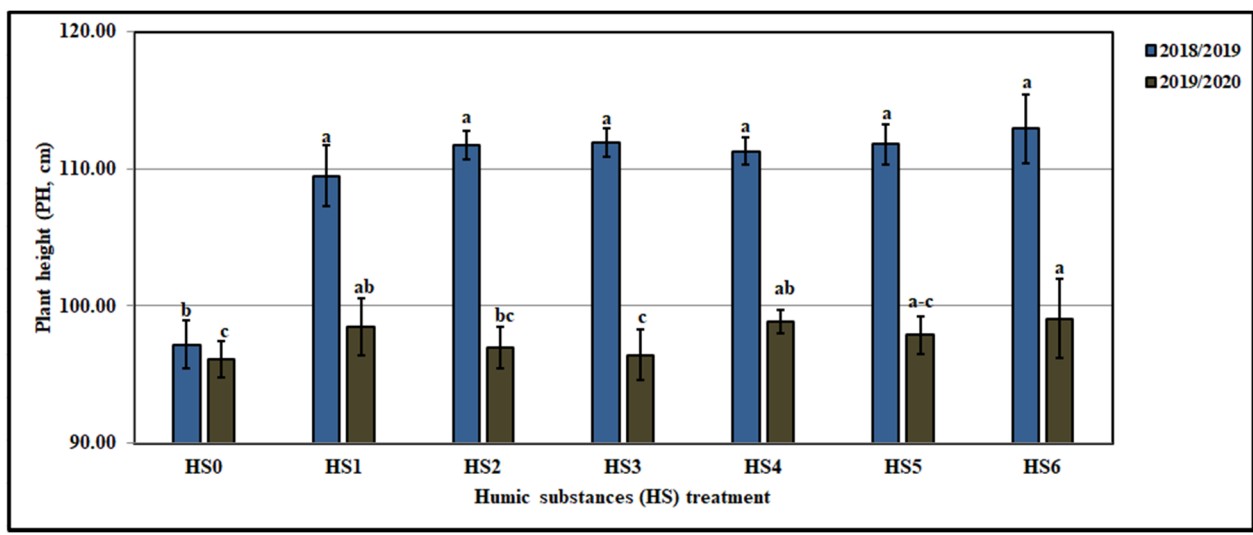

**Figure 1.** Influence of humic substances treatments on plant height (PH, cm) of some wheat varieties grown in clay loam soil in 2018/2019 and 2019/2020 seasons. Bars in the same years with a different letter indicate significant differences between treatments at $p \leq 0.01$. HS$_1$, HS$_2$, HS$_3$ represent HS applied as foliar spray at 1.0, 2.0, 4.0 g L$^{-1}$, respectively, HS$_4$, HS$_5$, HS$_6$ represent HS applied as soil application at 5.04, 7.56, 10.08 kg ha$^{-1}$.

### 3.1.2. Influence of Wheat Varieties

The results pertaining to the influence of the studied wheat varieties on PH, LA, and SPAD reading are graphically demonstrated in Figures 4–6. The results indicated that for PH, the varieties, in descending order, were ranked V$_3$ > V$_2$ > V$_1$ and V$_2$ > V$_3$ > V$_1$ in the first and second seasons. Similarly, the data indicated that for LA, the varieties, in descending order, were ranked V$_3$ > V$_1$ > V$_2$ in both seasons, and for SPAD reading, the varieties were ranked descending order as V$_1$ > V$_2$ > V$_3$ and V$_3$ > V$_1$ > V$_2$ in the two growth seasons, respectively. Analysis of variance showed that all varieties had a significant influence (at $p \leq 0.1$) on all studied parameters in both seasons, except for SPAD reading in the second season.

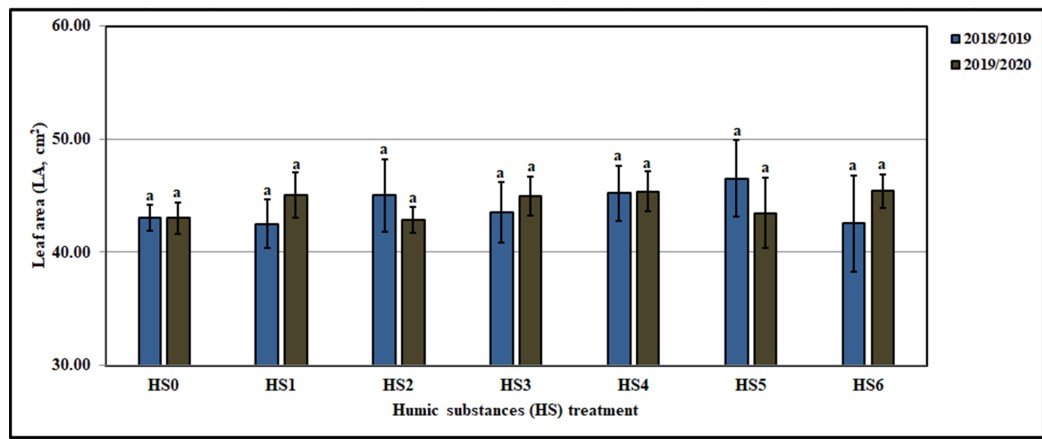

**Figure 2.** Influence of humic substances treatments on leaf area (LA, cm) of some wheat varieties grown in clay loam soil in 2018/2019 and 2019/2020 seasons. Bars in the same years with a different letter indicate significant differences between treatments at $p \leq 0.01$. $HS_1$, $HS_2$, $HS_3$ represent HS applied as foliar spray at 1.0, 2.0, 4.0 g $L^{-1}$, respectively, $HS_4$, $HS_5$, $HS_6$ represent HS applied as soil application at 5.04, 7.56, 10.08 kg $ha^{-1}$.

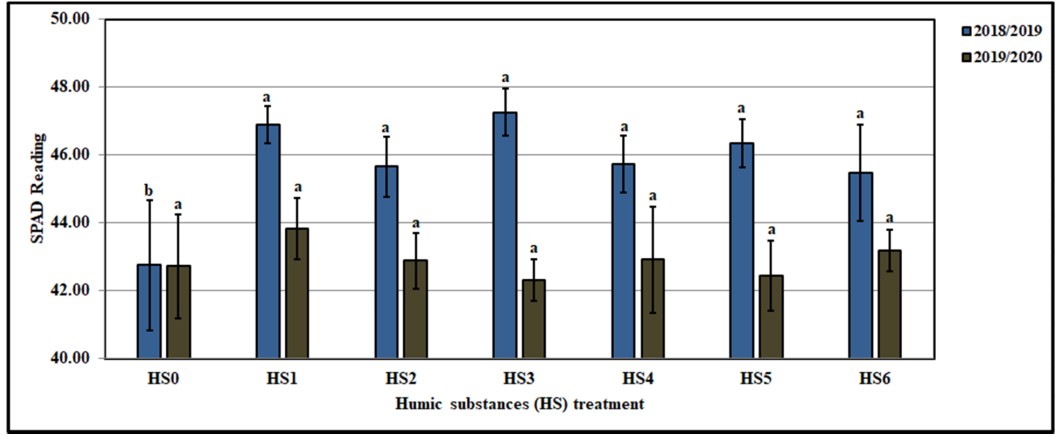

**Figure 3.** Influence of humic substances treatments on SPAD reading of some wheat varieties grown in clay loam soil in 2018/2019 and 2019/2020 seasons. Bars in the same years with a different letter indicate significant differences between treatments at $p \leq 0.01$. $HS_1$, $HS_2$, $HS_3$ represent HS applied as foliar spray at 1.0, 2.0, 4.0 g $L^{-1}$, respectively, $HS_4$, $HS_5$, $HS_6$ represent HS applied as soil application at 5.04, 7.56, 10.08 kg $ha^{-1}$.

### 3.1.3. Influence of the Humic Substance × Wheat Variety Interaction

According to the data in Table 5, the results revealed that all studied parameters improved as a result of the interaction between wheat varieties, and HS methods and doses, despite the analysis of variance indicating that all the studied wheat varieties had a non-significant effect on PH in 2018/2019 only, and on LA and SPAD reading in both seasons. On the other hand, varieties had highly significant effects on PH in the first season. Our investigation indicated that Giza168 and Seds1 plants fertilized with 5.04 kg $ha^{-1}$ ($V_3 \times HS_4$ and $V_2 \times HS_4$ treatments) produced the maximum values (121.10 and 102.74 cm, respectively) for PH in both seasons. However, Giza168 plants fertilized with 7.56 and 10.08 kg $ha^{-1}$ ($V_3 \times HS_5$ and $V_3 \times HS_6$) gave the maximum values (56.54 and 56.44 cm$^2$) for LA, and Seds1 plants nourished with 1.0 g $L^{-1}$ ($V_1 \times HS_3$ and $V_3 \times HS_4$ treatments) recorded the greatest mean values (49.12 and 44.59) for SPAD reading in both seasons. The results in Table 5 indicate that the percentages of increase compared with the lowest values were 37.98 and 16.03%, 93.10 and 53.83%, and 21.08 and 9.91% for PH, LA and SPAD reading during both seasons, respectively.

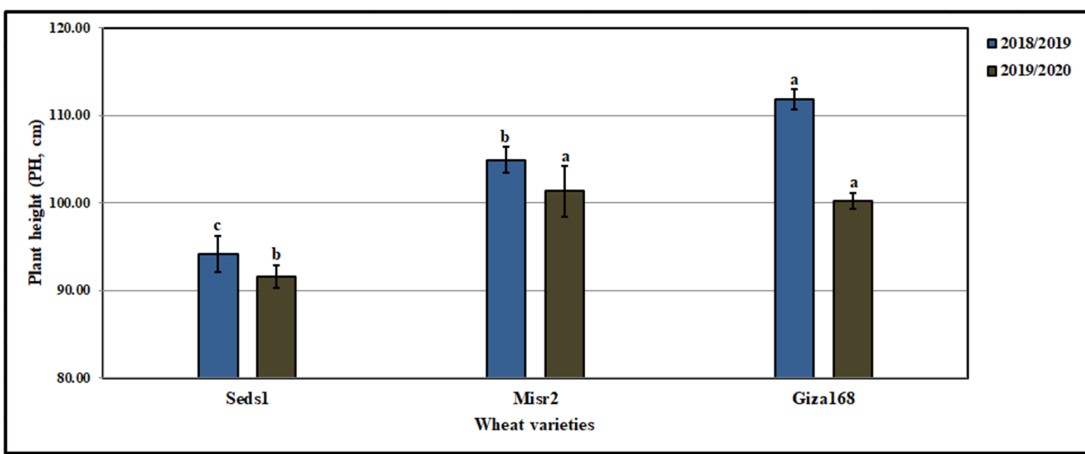

**Figure 4.** Influence of wheat varieties on plant height (PH, cm) in clay loam soil in 2018/2019 and 2019/2020 seasons. Bars in the same years with a different letter indicate significant differences between treatments at $p \leq 0.01$.

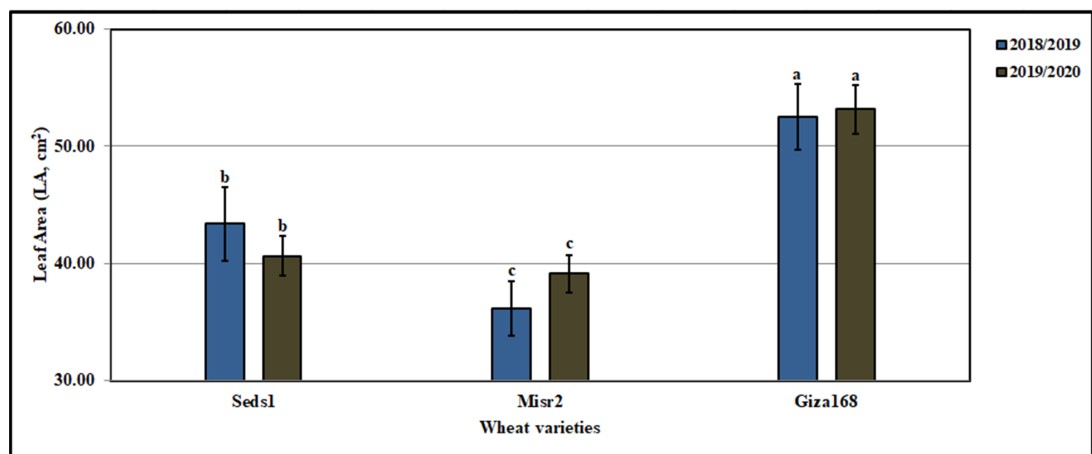

**Figure 5.** Influence of wheat varieties on leaf area (LA, cm$^2$) in clay loam soil in 2018/2019 and 2019/2020 seasons. Bars in the same years with a different letter indicate significant differences between treatments at $p \leq 0.01$.

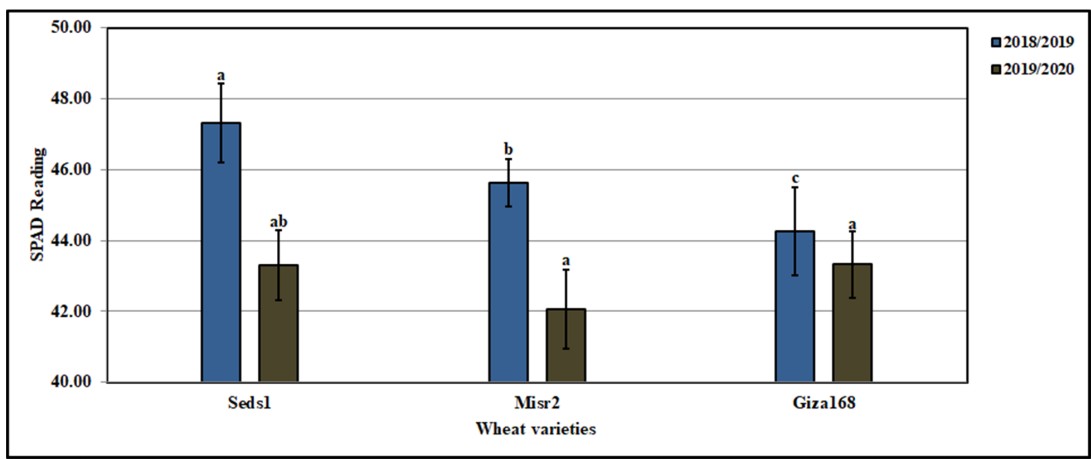

**Figure 6.** Influence of wheat varieties on SPAD reading in clay loam soil in 2018/2019 and 2019/2020 seasons. Bars in the same years with a different letter indicate significant differences between treatments at $p \leq 0.01$.

**Table 5.** Influence of interaction between humic substances and varieties on plant height, leaf area and SPAD reading of wheat plants grown in clay loam soil in 2018/2019 and 2019/2020 seasons.

| Treatment | (cm Plant$^{-1}$) | | | (cm$^2$) | | | SPAD Reading | | |
|---|---|---|---|---|---|---|---|---|---|
| | PH | | | LA | | | | | |
| HS | V$_1$ | V$_2$ | V$_3$ | V$_1$ | V$_2$ | V$_3$ | V$_1$ | V$_2$ | V$_3$ |
| | | | | | **2018/2019 season** | | | | |
| HS$_0$ | 87.80 $^d$ ± 1.7 | 100.86 $^c$ ± 1.8 | 99.76 $^c$ ± 1.7 | 42.68 $^{de}$ ± 1.1 | 39.26 $^{ef}$ ± 1.2 | 47.08 $^{a-e}$ ± 1.2 | 44.55 $^{de}$ ± 1.7 | 40.57 $^f$ ± 1.7 | 43.05 $^{ef}$ ± 2.3 |
| HS$_1$ | 99.31 $^c$ ± 2.3 | 112.89 $^b$ ± 3.4 | 118.51 $^{ab}$ ± 1.1 | 39.50 $^{ef}$ ± 0.2 | 36.82 $^{ef}$ ± 3.7 | 50.83 $^{a-d}$ ± 2.5 | 47.85 $^{ab}$ ± 0.4 | 47.59 $^{a-c}$ ± 0.6 | 45.26 $^{b-e}$ ± 0.7 |
| HS$_2$ | 100.71 $^c$ ± 1.5 | 113.77 $^b$ ± 1.1 | 117.84 $^{ab}$ ± 0.5 | 45.75 $^{b-e}$ ± 4.0 | 36.96 $^{ef}$ ± 3.0 | 52.35 $^{a-d}$ ± 2.6 | 45.53 $^{b-e}$ ± 1.7 | 46.81 $^{a-d}$ ± 0.2 | 44.61 $^{c-e}$ ± 0.7 |
| HS$_3$ | 100.08 $^c$ ± 2.1 | 113.47 $^b$ ± 0.4 | 118.37 $^{ab}$ ± 0.5 | 43.07 $^{c-e}$ ± 1.7 | 36.87 $^{ef}$ ± 2.7 | 50.54 $^{a-d}$ ± 3.6 | 49.12 $^a$ ± 0.6 | 47.02 $^{a-d}$ ± 0.7 | 45.66 $^{b-e}$ ± 1.2 |
| HS$_4$ | 100.20 $^c$ ± 0.9 | 114.13 $^b$ ± 1.1 | 113.49 $^b$ ± 1.2 | 44.94 $^{c-e}$ ± 4.3 | 36.48 $^{ef}$ ± 1.67 | 54.30 $^{a-c}$ ± 1.3 | 47.62 $^{a-c}$ ± 1.0 | 45.93 $^{b-e}$ ± 0.7 | 43.63 $^e$ ± 0.87 |
| HS$_5$ | 101.42 $^c$ ± 1.3 | 114.89 $^{ab}$ ± 1.3 | 121.15 $^a$ ± 1.8 | 45.49 $^{b-e}$ ± 4.8 | 37.51 $^{ef}$ ± 1.27 | 56.54 $^a$ ± 4.07 | 49.11 $^a$ ± 0.5 | 46.81 $^{a-d}$ ± 0.2 | 43.12 $^{ef}$ ± 1.5 |
| HS$_6$ | 98.68 $^c$ ± 4.7 | 112.11 $^b$ ± 1.3 | 113.17 $^b$ ± 1.6 | 42.48 $^{de}$ ± 5.7 | 29.28 $^f$ ± 2.80 | 55.88 $^{ab}$ ± 4.33 | 47.34 $^{a-d}$ ± 2.0 | 44.61 $^{c-e}$ ± 1.0 | 44.48 $^{de}$ ± 1.3 |
| | | | | | **2018/2019 season** | | | | |
| HS$_0$ | 87.80 $^f$ ± 1.73 | 100.86 $^{ab}$ ± 1.13 | 99.76 $^{ab}$ ± 1.17 | 42.68 $^{cd}$ ± 1.15 | 39.26 $^d$ ± 1.53 | 47.08 $^{bc}$ ± 1.53 | 44.59 $^a$ ± 2.3 | 40.57 $^c$ ± 1.2 | 43.05 $^{a-c}$ ± 1.2 |
| HS$_1$ | 92.61 $^{cd}$ ± 0.8 | 101.12 $^{ab}$ ± 4.1 | 101.80 $^{ab}$ ± 2.3 | 41.00 $^d$ ± 2.5 | 40.95 $^d$ ± 1.7 | 53.15 $^a$ ± 1.7 | 44.27 $^{ab}$ ± 0.7 | 43.63 $^{a-c}$ ± 1.0 | 43.57 $^{a-c}$ ± 0.9 |
| HS$_2$ | 91.45 $^{cd}$ ± 1.0 | 100.65 $^{ab}$ ± 2.9 | 98.89 $^b$ ± 0.7 | 40.49 $^d$ ± 1.2 | 36.69 $^d$ ± 1.3 | 51.47 $^{ab}$ ± 0.9 | 43.19 $^{a-c}$ ± 0.2 | 42.35 $^{a-c}$ ± 1.5 | 43.10 $^{a-c}$ ± 0.7 |
| HS$_3$ | 90.01 $^{de}$ ± 0.6 | 100.05 $^{ab}$ ± 3.9 | 99.29 $^{ab}$ ± 1.9 | 39.47 $^d$ ± 0.8 | 40.13 $^d$ ± 2.5 | 55.30 $^a$ ± 1.9 | 42.72 $^{a-c}$ ± 0.4 | 41.63 $^{a-c}$ ± 0.8 | 42.60 $^{a-c}$ ± 0.6 |
| HS$_4$ | 91.97 $^{cd}$ ± 0.4 | 102.74 $^a$ ± 0.9 | 101.84 $^{ab}$ ± 1.3 | 40.84 $^d$ ± 1.7 | 39.44 $^d$ ± 1.2 | 55.84 $^a$ ± 2.5 | 42.47 $^{a-c}$ ± 1.2 | 41.66 $^{a-c}$ ± 1.6 | 44.59 $^a$ ± 1.9 |
| HS$_5$ | 92.91 $^{cd}$ ± 1.8 | 101.87 $^{ab}$ ± 2.0 | 98.86 $^b$ ± 0.4 | 40.39 $^d$ ± 2.8 | 37.37 $^d$ ± 1.9 | 52.68 $^a$ ± 4.5 | 43.29 $^{a-c}$ ± 1.1 | 41.12 $^{a-c}$ ± 1.5 | 42.93 $^{a-c}$ ± 0.5 |
| HS$_6$ | 94.35 $^c$ ± 2.8 | 102.01 $^{ab}$ ± 5.5 | 100.88 $^{ab}$ ± 0.4 | 39.63 $^d$ ± 1.9 | 40.12 $^d$ ± 0.8 | 56.44 $^a$ ± 1.7 | 42.64 $^{a-c}$ ± 0.9 | 43.50 $^{a-c}$ ± 0.2 | 43.32 $^a$ ± 0.7 |

Means in the same column denoted by a different letter indicate significant difference between treatments at ≤0.05. HS$_1$, HS$_2$, HS$_3$ represent HS applied as foliar spray at 1.0, 2.0, 4.0 g L$^{-1}$, respectively, HS$_4$, HS$_5$, HS$_6$ represent HS applied as soil application at 5.04, 7.56, 10.08 kg ha$^{-1}$. V$_1$ = Seds1 cv, V$_2$ = Misr2 cv, V$_3$ = Giza168 cv. PH = Plant height, LA = Leaf area.

### 3.2. Yield and Its Components

3.2.1. Influence of Humic Substances, Wheat Varieties and Their Interactions on Spike Length, Grain Weight, and Grain Number

Influence of Humic Substances

Analysis of variance showed a significant influence (at $p \leq 0.1$) on grain weight per spike (GWS) and grain number per spike (GNS) in the first season only. On the contrary, spike length (SpL) in both growth seasons, as well as GWS and GNS, showed non-significant impacts in the second season only. The results of our field study, as presented in Figures 7–9, indicated that the highest values (11.62 and 11.44 cm) were recorded by the $HS_1$ and $HS_0$ treatments for SpL. In addition, the highest values for GWS (2.93 and 2.43 g spike$^{-1}$) were recorded for the $HS_1$ and $HS_4$ treatments, and the highest values for GNS (69.49 and 56.03) were found in plants treated with $HS_5$ and $HS_4$ in 2018/2019 and 2019/2020, respectively. Based on the comparison between the highest and lowest values, the percentages of increase reached 4.97 and 5.05%, 30.22 and 12.50%, and 28.77 and 7.07% for all the aforementioned attributes in the two seasons.

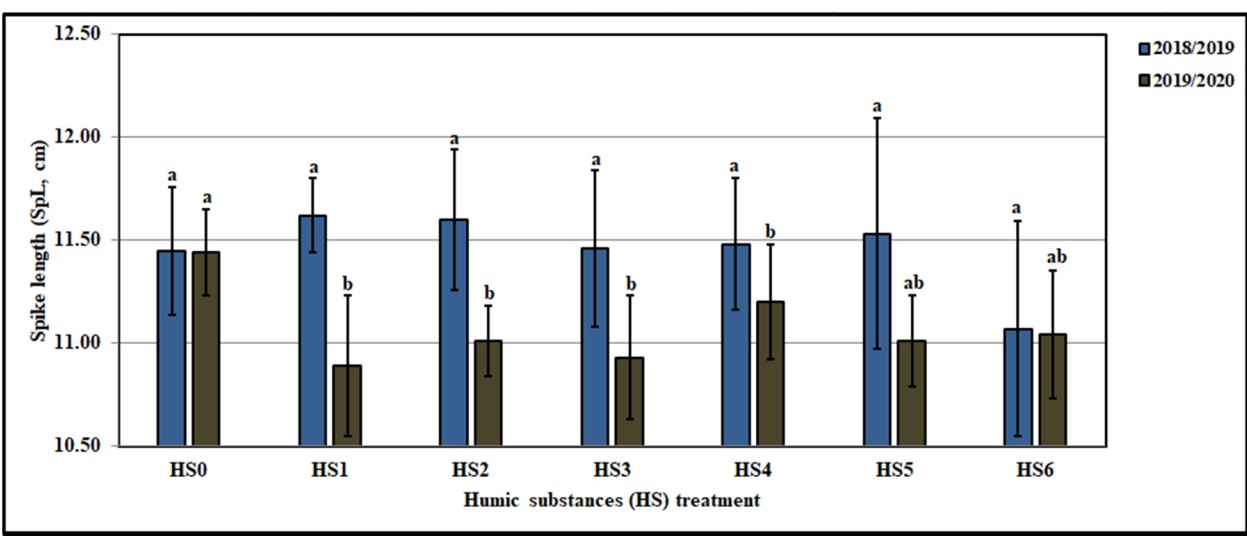

**Figure 7.** Influence of humic substances treatments on spike length (SpL, cm) of some wheat varieties grown in clay loam soil in 2018/2019 and 2019/2020 seasons. Bars in the same years with a different letter indicate significant differences between treatments at $p \leq 0.01$. $HS_1$, $HS_2$, $HS_3$ represent HS applied as foliar spray at 1.0, 2.0, 4.0 g L$^{-1}$, respectively, $HS_4$, $HS_5$, $HS_6$ represent HS applied as soil application at 5.04, 7.56, 10.08 kg ha$^{-1}$.

Influence of Wheat Varieties

The results presented in Figures 10–12 indicated that cultivar had a marked impact on SpL, GWS and GNS in the first season as compared with the second season. However, the best values (12.19 and 11.62 cm, 11.67 and 11.52 g spike$^{-1}$, and 10.51 and 10.37 n spike$^{-1}$) were produced by $V_3$, followed by $V_1$ then $V_2$ in both growth seasons. Values of 15.99 and 12.05% were recorded as the increase in percentage for SpL compared with the minimum values in 2018–2019 and 2019–2020, respectively. The results obtained from the statistical analysis revealed highly significant differences among the studied varieties in both growth seasons, respectively. Dissimilar data were found for GWS and GNS; however, the varieties were ranked in descending order as $V_3 > V_2 > V_1$ (2.85 > 2.70 > 2.51) and $V_2 > V_1 > V_3$ (2.40 > 2.29 > 2.24) for GWS in 2018/2019 and 2019/2020, respectively, and as $V_1 > V_2 > V_3$ (67.74 > 66.05 > 62.61) and $V_2 > V_1 > V_3$ (58.27 > 55.07 > 49.17) for GNS in 2018/2019 and 2019/2020, respectively. It was noticed that the increasing percentages in comparison with the lowest values amounted to 13.55 and 7.14%, and 8.19 and 18.51% for GWS and GNS, respectively, in both growth seasons. The statistical analysis showed highly significant

effects of variety on GWS and GNS in the first season and a non-significant effect on both attributes in the second season.

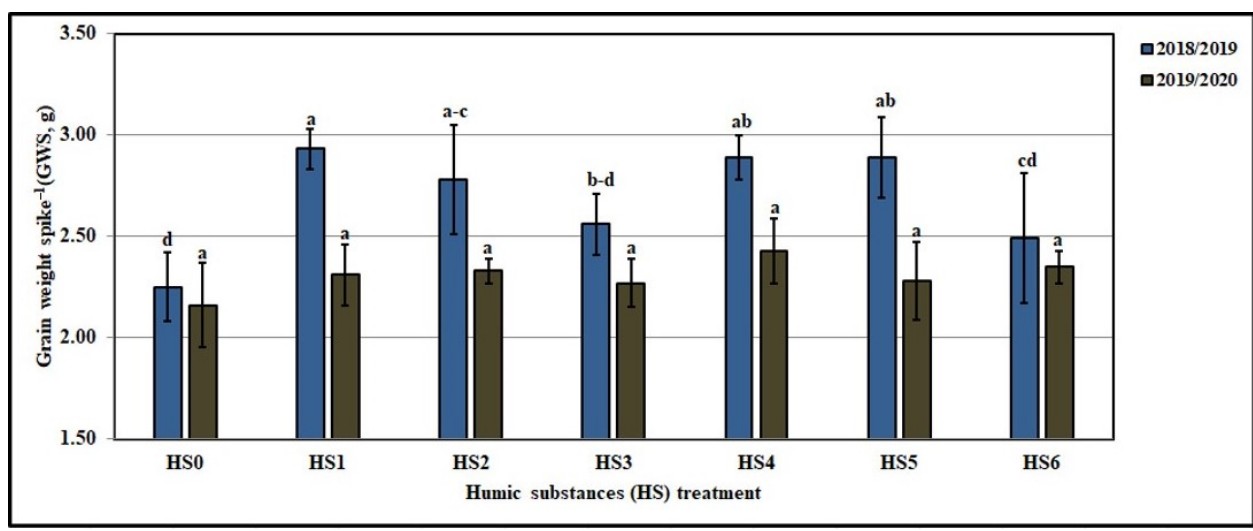

**Figure 8.** Influence of humic substances treatments on grain weight per spike (GWS, g) of some wheat varieties grown in clay loam soil in 2018/2019 and 2019/2020 seasons. Bars in the same years with a different letter indicate significant differences between treatments at $p \leq 0.01$. $HS_1$, $HS_2$, $HS_3$ represent HS applied as foliar spray at 1.0, 2.0, 4.0 g $L^{-1}$, respectively, $HS_4$, $HS_5$, $HS_6$ represent HS applied as soil application at 5.04, 7.56, 10.08 kg $ha^{-1}$.

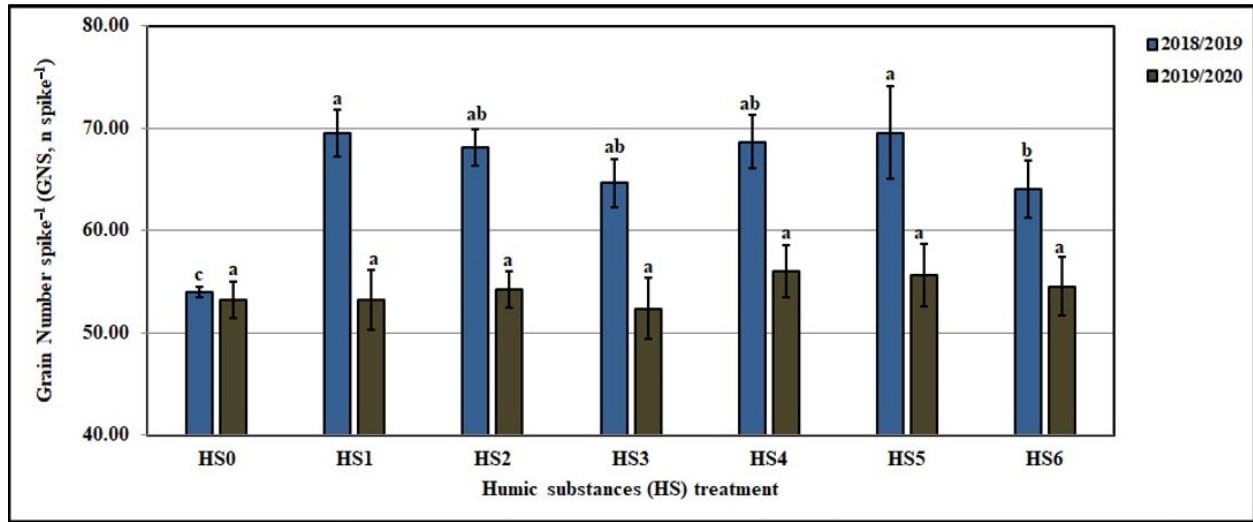

**Figure 9.** Influence of humic substances treatments on grain number per spike (GNS, n $spike^{-1}$) of some wheat varieties grown in clay loam soil in 2018/2019 and 2019/2020 seasons. Bars in the same years with a different letter indicate significant differences between treatments at $p \leq 0.01$. $HS_1$, $HS_2$, $HS_3$ represent HS applied as foliar spray at 1.0, 2.0, 4.0 g $L^{-1}$, respectively, $HS_4$, $HS_5$, $HS_6$ represent HS applied as soil application at 5.04, 7.56, 10.08 kg $ha^{-1}$.

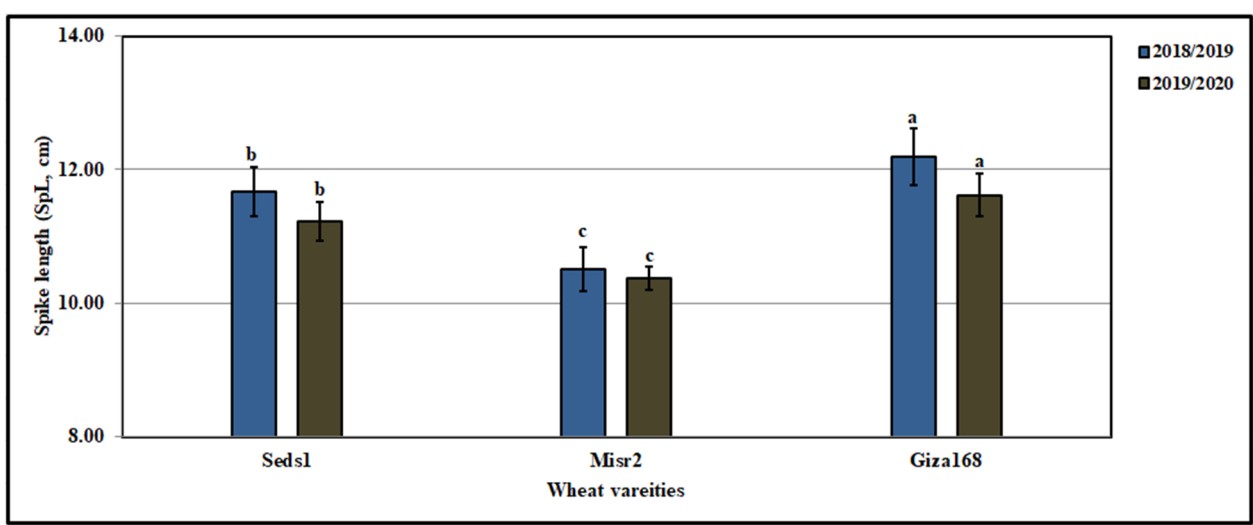

**Figure 10.** Influence of wheat varieties on spike length (SpL, cm) in clay loam soil in 2018/2019 and 2019/2020 seasons. Bars in the same years with a different letter indicate significant differences between treatments at $p \leq 0.01$. $HS_1$, $HS_2$, $HS_3$ represent HS applied as foliar spray at 1.0, 2.0, 4.0 g $L^{-1}$, respectively, $HS_4$, $HS_5$, $HS_6$ represent HS applied as soil application at 5.04, 7.56, 10.08 kg $ha^{-1}$.

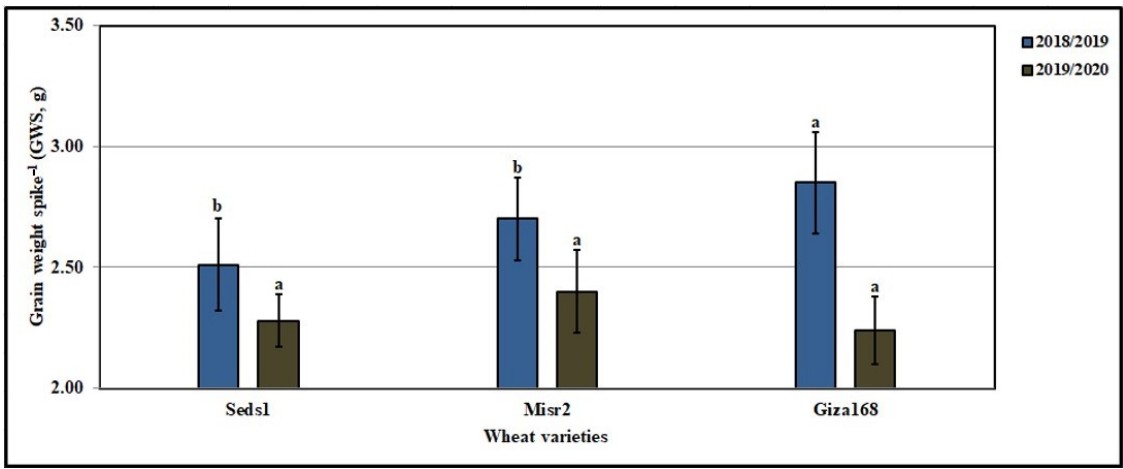

**Figure 11.** Influence of wheat varieties on grain weight $spike^{-1}$ (GWS, g) in clay loam soil in 2018/2019 and 2019/2020 seasons. Bars in the same years with a different letter indicate significant differences between treatments at $p \leq 0.01$. $HS_1$, $HS_2$, $HS_3$ represent HS applied as foliar spray at 1.0, 2.0, 4.0 g $L^{-1}$, respectively, $HS_4$, $HS_5$, $HS_6$ represent HS applied as soil application at 5.04, 7.56, 10.08 kg $ha^{-1}$.

Influence of the Humic Substance × Wheat Variety Interaction

Regarding the influence of the HS × wheat variety interaction on SpL, GWS and GNS, the data in Table 6 revealed that the greatest values of SpL (12.78 and 12.30 cm) were recorded for $V_3 \times HS_5$ and untreated Seds1 plants ($V_1 \times HS_0$), respectively, in both growth seasons. Similarly, the maximum mean values for GWS (3.13 and 2.53 g) were achieved by the $V_3 \times HS_5$ and $V_3 \times HS_4$ treatments, and the maximum values for GNS (72.79 and 62.36) were achieved by Seds1 plants nourished with 1.0 g $L^{-1}$ of HS ($V_1 \times HS_1$) and Misr2 plants sprayed with 2.0 g $L^{-1}$ of HS ($V_2 \times HS_2$), respectively, in both seasons. As depicted in Table 6, the rates of increase compared with the lowest values were 27.67 and 19.77%, 52.68 and 47.09% and 53.79 and 38.12% for SpL, GWS and GNS in both seasons, respectively. The results of the ANOVA indicated that all the aforementioned attributes had no significant influences.

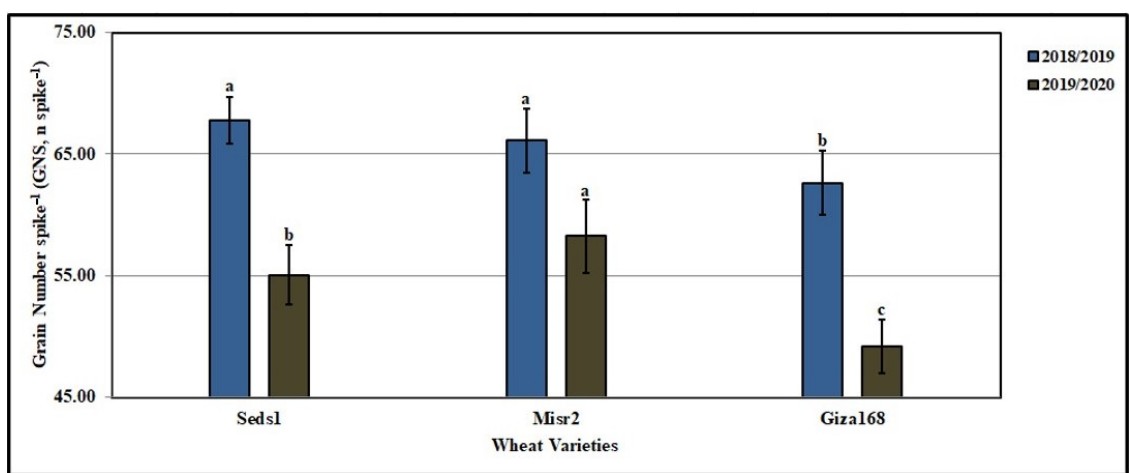

**Figure 12.** Influence of wheat varieties on grain number spike$^{-1}$ (GNS, n spike$^{-1}$) in clay loam soil in 2018/2019 and 2019/2020 seasons. Bars in the same years with a different letter indicate significant differences between treatments at $p \leq 0.01$. HS$_1$, HS$_2$, HS$_3$ represent HS applied as foliar spray at 1.0, 2.0, 4.0 g L$^{-1}$, respectively, HS$_4$, HS$_5$, HS$_6$ represent HS applied as soil application at 5.04, 7.56, 10.08 kg ha$^{-1}$.

3.2.2. Influence of Humic Substances, Wheat Varieties and Their Interaction on 1000-Grain Weight, Biological Yield and Total Grain Yield

Influence of Humic Substances

The results indicated that HS, either as a foliar spray or a soil application, significantly (at $p \leq 0.01$) affected biological yield (BY) in the second season and total grain yield (TGY) in the first season. However, HS had no significant effect on the 1000-grain weight (TGW). As shown graphically (Figures 13–15), the greatest values for TGW, BY and TGY were 44.09 and 43.32 g, 4.91 and 3.68 kg, and 6.60 and 7.80 t ha$^{-1}$ in the 2018/2019 and 2019/2020 seasons, respectively. These results indicated that the rates of increase reached 9.24 and 8.30%, 67.58 and 30.04%, and 75.85 and 72.02% for TGW, BY and TGY in 2018/2019 and 2019/2020, respectively. Similarly, the general trend of the data portrayed in Figures 13–15 reveal that all the above-mentioned parameters were higher in the first season than in the second season.

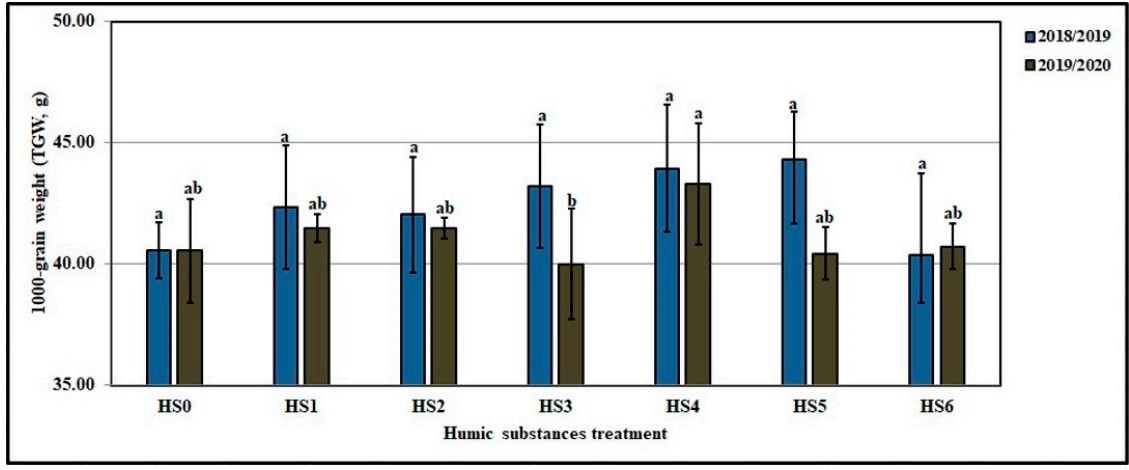

**Figure 13.** Influence of humic substances treatments on 1000-grain weight (TGW, g) of some wheat varieties grown in clay loam soil in 2018/2019 and 2019/2020 seasons. Bars in the same years with a different letter indicate significant differences between treatments at $p \leq 0.01$. HS$_1$, HS$_2$, HS$_3$ represent HS applied as foliar spray at 1.0, 2.0, 4.0 g L$^{-1}$, respectively, HS$_4$, HS$_5$, HS$_6$ represent HS applied as soil application at 5.04, 7.56, 10.08 kg ha$^{-1}$.

**Table 6.** Influence of the interaction between humic substances and varieties on spike length, grain weight and grain number per spike of wheat plants grown in clay loam soil in 2018/2019 and 2019/2020 seasons.

| Treatment | (cm Plant$^{-1}$) | | | (g Spike$^{-1}$) | | | (n Spike$^{-1}$) | | |
|---|---|---|---|---|---|---|---|---|---|
| | SpL | | | GWS | | | GNS | | |
| HS | V$_1$ | V$_2$ | V$_3$ | V$_1$ | V$_2$ | V$_3$ | V$_1$ | V$_2$ | V$_3$ |
| 2018/2019 season | | | | | | | | | |
| HS$_0$ | 12.30 $^{ab}$ ± 0.18 | 10.34 $^{f-h}$ ± 0.17 | 11.72 $^{a-e}$ ± 0.58 | 2.27 $^{gh}$ ± 0.07 | 2.50 $^{c-h}$ ± 0.14 | 2.05 $^{h}$ ± 0.29 | 55.91 $^{e}$ ± 0.58 | 57.45 $^{de}$ ± 0.33 | 46.25 $^{f}$ ± 0.65 |
| HS$_1$ | 12.07 $^{a-d}$ ± 0.12 | 10.67 $^{e-h}$ ± 0.29 | 12.13 $^{a-d}$ ± 0.12 | 2.87 $^{a-f}$ ± 0.10 | 2.92 $^{a-e}$ ± 0.03 | 3.01 $^{a-c}$ ± 0.17 | 72.79 $^{a}$ ± 1.29 | 70.49 $^{ab}$ ± 2.22 | 65.19 $^{a-d}$ ± 3.22 |
| HS$_2$ | 11.50 $^{a-f}$ ± 0.61 | 10.73 $^{e-h}$ ± 0.22 | 12.57 $^{ab}$ ± 0.20 | 2.41 $^{d-h}$ ± 0.37 | 2.84 $^{a-f}$ ± 0.22 | 3.08 $^{ab}$ ± 0.23 | 68.42 $^{a-c}$ ± 0.71 | 68.89 $^{ab}$ ± 1.31 | 66.82 $^{a-c}$ ± 3.31 |
| HS$_3$ | 11.37 $^{b-g}$ ± 0.60 | 10.85 $^{c-h}$ ± 0.48 | 12.17 $^{a-c}$ ± 0.07 | 2.39 $^{e-h}$ ± 0.13 | 2.36 $^{f-h}$ ± 0.21 | 2.94 $^{a-d}$ ± 0.10 | 69.66 $^{a-c}$ ± 1.60 | 63.15 $^{b-e}$ ± 2.24 | 61.05 $^{c-e}$ ± 3.13 |
| HS$_4$ | 11.47 $^{a-g}$ ± 0.26 | 10.83 $^{d-h}$ ± 0.37 | 12.15 $^{a-d}$ ± 0.33 | 2.69 $^{a-g}$ ± 0.06 | 3.04 $^{ab}$ ± 0.14 | 2.93 $^{a-d}$ ± 0.13 | 69.99 $^{ab}$ ± 4.03 | 72.76 $^{a}$ ± 1.67 | 63.22 $^{a-e}$ ± 2.05 |
| HS$_5$ | 11.63 $^{a-f}$ ± 0.64 | 10.17 $^{gh}$ ± 0.42 | 12.78 $^{a}$ ± 0.64 | 2.89 $^{a-e}$ ± 0.25 | 2.65 $^{a-g}$ ± 0.23 | 3.13 $^{a}$ ± 0.14 | 72.39 $^{a}$ ± 3.74 | 65.63 $^{a-d}$ ± 5.95 | 70.65 $^{ab}$ ± 3.95 |
| HS$_6$ | 11.38 $^{b-g}$ ± 0.18 | 10.01 $^{h}$ ± 0.37 | 11.82 $^{a-e}$ ± 1.02 | 2.09 $^{h}$ ± 0.33 | 2.57 $^{b-h}$ ± 0.22 | 2.81 $^{a-f}$ ± 0.39 | 64.02 $^{a-e}$ ± 1.43 | 63.98 $^{a-e}$ ± 4.92 | 64.02 $^{a-e}$ ± 2.15 |
| 2018/2019 season | | | | | | | | | |
| HS$_0$ | 12.28 $^{a}$ ± 0.17 | 10.29 $^{e}$ ± 0.17 | 11.77 $^{a-c}$ ± 0.28 | 2.25 $^{ab}$ ± 0.12 | 2.45 $^{ab}$ ± 0.23 | 2.02 $^{b}$ ± 0.28 | 55.89 $^{a-e}$ ± 1.73 | 57.41 $^{a-d}$ ± 2.31 | 46.18 $^{f}$ ± 1.15 |
| HS$_1$ | 11.02 $^{c-e}$ ± 0.45 | 10.51 $^{de}$ ± 0.11 | 11.15 $^{c-e}$ ± 0.45 | 2.21 $^{ab}$ ± 0.02 | 2.50 $^{ab}$ ± 0.19 | 2.23 $^{ab}$ ± 0.23 | 54.33 $^{a-e}$ ± 2.44 | 60.06 $^{ab}$ ± 2.62 | 45.15 $^{f}$ ± 3.77 |
| HS$_2$ | 10.91 $^{c-e}$ ± 0.12 | 10.43 $^{e}$ ± 0.13 | 11.69 $^{a-c}$ ± 0.26 | 2.22 $^{ab}$ ± 0.05 | 2.49 $^{ab}$ ± 0.06 | 2.27 $^{ab}$ ± 0.07 | 50.53 $^{c-f}$ ± 1.27 | 62.36 $^{a}$ ± 2.83 | 49.81 $^{d-f}$ ± 1.34 |
| HS$_3$ | 11.04 $^{c-e}$ ± 0.52 | 10.27 $^{e}$ ± 0.13 | 11.46 $^{a-c}$ ± 0.25 | 2.37 $^{ab}$ ± 0.14 | 2.17 $^{ab}$ ± 0.14 | 2.27 $^{ab}$ ± 0.07 | 55.45 $^{a-e}$ ± 3.61 | 53.00 $^{b-f}$ ± 4.32 | 48.55 $^{ef}$ ± 1.03 |
| HS$_4$ | 11.07 $^{c-e}$ ± 0.17 | 10.40 $^{e}$ ± 0.27 | 12.13 $^{ab}$ ± 0.39 | 2.32 $^{ab}$ ± 0.14 | 2.45 $^{ab}$ ± 0.20 | 2.53 $^{a}$ ± 0.15 | 55.16 $^{a-e}$ ± 2.50 | 48.16 $^{a-c}$ ± 2.10 | 54.76 $^{a-e}$ ± 3.07 |
| HS$_5$ | 10.95 $^{c-e}$ ± 0.31 | 10.34 $^{e}$ ± 0.18 | 11.74 $^{a-c}$ ± 0.16 | 2.28 $^{ab}$ ± 0.19 | 2.22 $^{ab}$ ± 0.27 | 2.35 $^{ab}$ ± 0.10 | 57.15 $^{a-d}$ ± 3.22 | 58.56 $^{a-c}$ ± 3.27 | 51.34 $^{c-f}$ ± 2.59 |
| HS$_6$ | 11.34 $^{bc}$ ± 0.31 | 10.31 $^{e}$ ± 0.21 | 11.47 $^{a-c}$ ± 0.43 | 2.29 $^{ab}$ ± 0.07 | 2.44 $^{ab}$ ± 0.08 | 2.32 $^{ab}$ ± 0.09 | 57.92 $^{a-d}$ ± 2.54 | 58.27 $^{a-c}$ ± 3.74 | 48.44 $^{ef}$ ± 2.43 |

Means in the same column denoted by a different letter indicate significant difference between treatments at ≤0.05. HS$_1$, HS$_2$, HS$_3$ represent HS applied as foliar spray at 1.0, 2.0, 4.0 g L$^{-1}$, respectively, HS$_4$, HS$_5$, HS$_6$ represent HS applied as soil application at 5.04, 7.56, 10.08 kg ha$^{-1}$. V$_1$ = Seds1 cv, V$_2$ = Misr2 cv, V$_3$ = Giza168 cv, SpL = Spike length, GWS = Grain weight spike$^{-1}$, GNS = Grain number spike$^{-1}$.

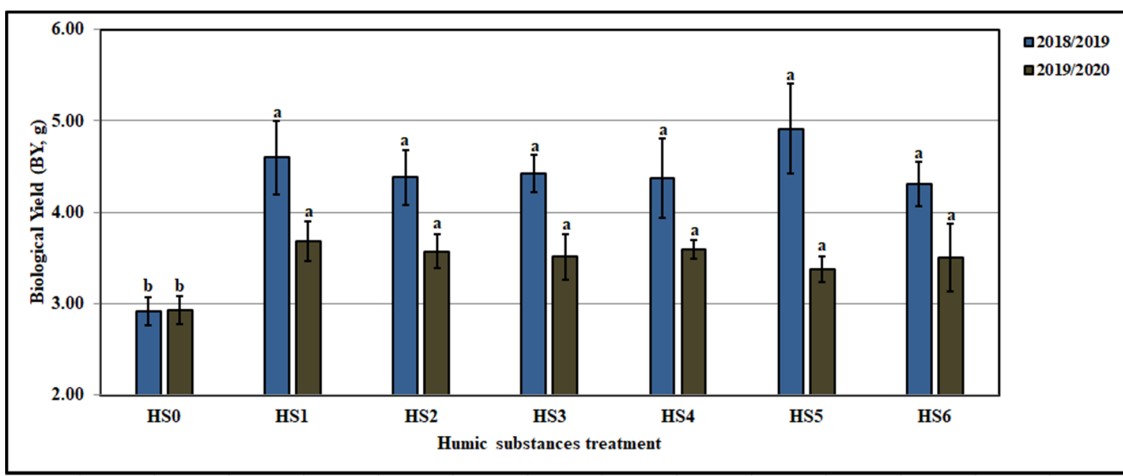

**Figure 14.** Influence of humic substances treatments on biological yield (BY, g plant$^{-1}$) of some wheat varieties grown in clay loam soil in 2018/2019 and 2019/2020 seasons. Bars in the same years with a different letter indicate significant differences between treatments at $p \leq 0.01$. HS$_1$, HS$_2$, HS$_3$ represent HS applied as foliar spray at 1.0, 2.0, 4.0 g L$^{-1}$, respectively, HS$_4$, HS$_5$, HS$_6$ represent HS applied as soil application at 5.04, 7.56, 10.08 kg ha$^{-1}$.

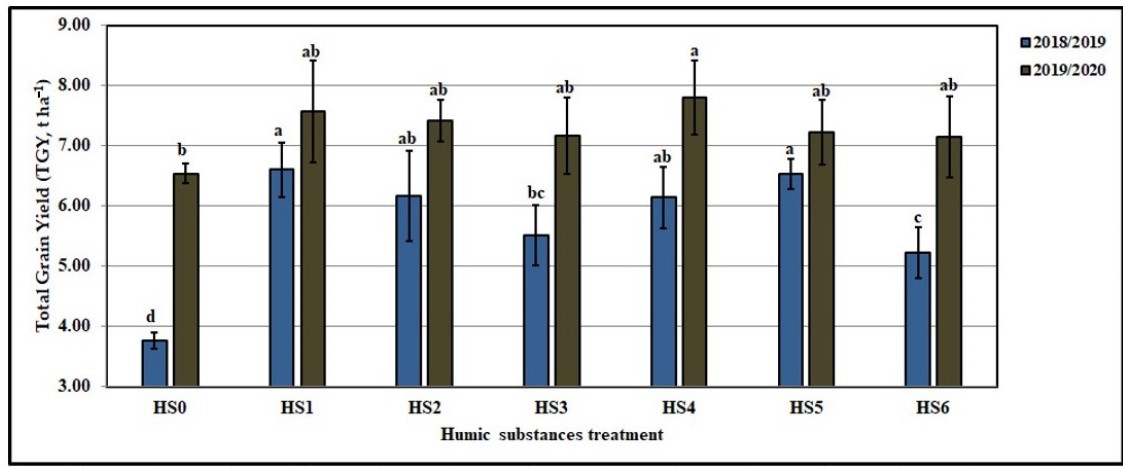

**Figure 15.** Influence of humic substances treatments on total grain yield (TGY, t ha$^{-1}$) of some wheat varieties grown in clay loam soil in 2018/2019 and 2019/2020 seasons. Bars in the same years with a different letter indicate significant differences between treatments at $p \leq 0.01$. HS$_1$, HS$_2$, HS$_3$ represent HS applied as foliar spray at 1.0, 2.0, 4.0 g L$^{-1}$, respectively, HS$_4$, HS$_5$, HS$_6$ represent HS applied as soil application at 5.04, 7.56, 10.08 kg ha$^{-1}$.

Influence of Wheat Varieties

The analysis of variance showed a significant (at $p \leq 0.01$) effect on TGW and BY in both seasons and TGY in the first season only, but a non-significant effect on TGY in the second season. The results in Figures 16–18 show that the varieties were ranked in descending order as V$_3$ > V$_2$ > V$_1$ (47.51 > 40.32 > 39.75 g) and V$_3$ > V$_1$ > V$_2$ (45.57 > 38.97 > 38.33 g) for TGW, and V$_2$ > V$_3$ > V$_1$ (4.43 > 4.30 > 4.10 kg plant$^{-1}$) and V$_3$ > V$_2$ > V$_1$ (3.78 > 3.29 > 3.28 kg plant$^{-1}$) for BY in 2018/2019 and 2019/2020, respectively. With regard to TGY, the studied varieties could be arranged in the following order: V$_3$ > V$_2$ > V$_1$ (6.00 > 5.87 > 5.26 and 7.48 > 7.33 > 6.99 ton ha$^{-1}$) in both growth seasons. According to the highest and lowest values obtained, the percentages of increase were 19.52 and 18.89%, 15.24 and 13.16%, and 14.17 and 6.89% for TGW, BY and TGY in the 2018/2019 and 2019/2020 seasons, respectively.

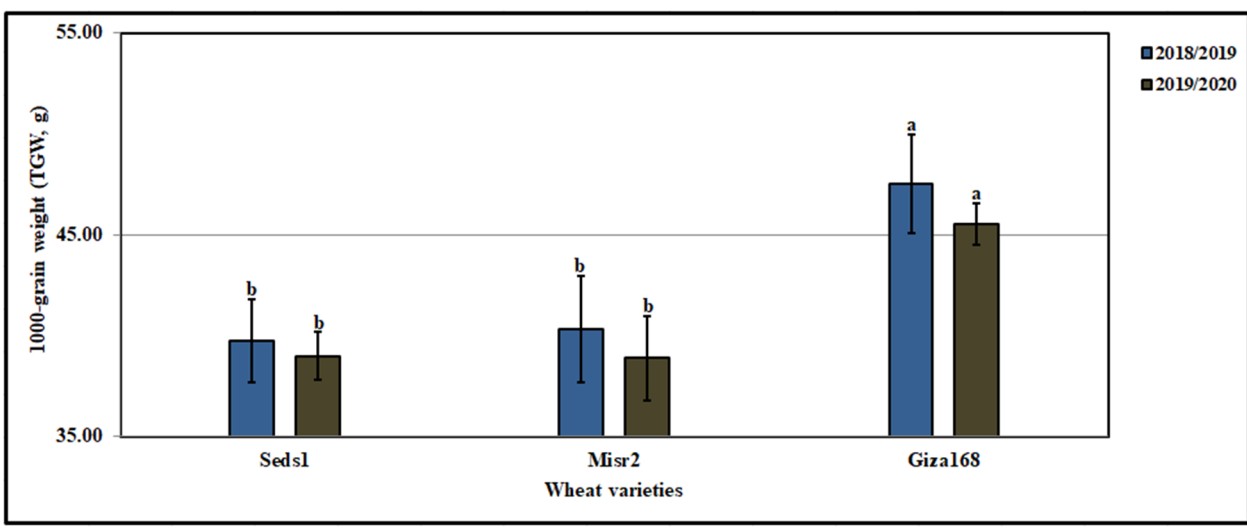

**Figure 16.** Influence of wheat varieties on 1000-grain weight (TGW, g) in clay loam soil in 2018/2019 and 2019/2020 seasons. Bars in the same years with a different letter indicate significant differences between treatments at $p \leq 0.01$. $HS_1$, $HS_2$, $HS_3$ represent HS applied as foliar spray at 1.0, 2.0, 4.0 g L$^{-1}$, respectively, $HS_4$, $HS_5$, $HS_6$ represent HS applied as soil application at 5.04, 7.56, 10.08 kg ha$^{-1}$.

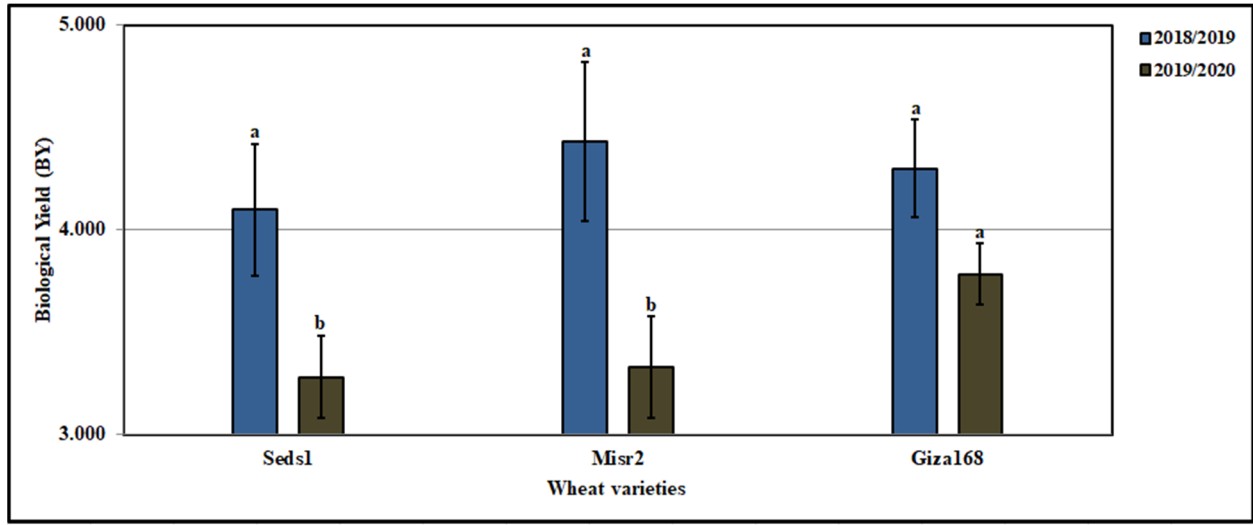

**Figure 17.** Influence of wheat varieties on biological yield (BY, kg) in clay loam soil in 2018/2019 and 2019/2020 seasons. Bars in the same years with a different letter indicate significant differences between treatments at $p \leq 0.01$. $HS_1$, $HS_2$, $HS_3$ represent HS applied as foliar spray at 1.0, 2.0, 4.0 g L$^{-1}$, respectively, $HS_4$, $HS_5$, $HS_6$ represent HS applied as soil application at 5.04, 7.56, 10.08 kg ha$^{-1}$.

Influence of the Humic Substance × Wheat Variety Interaction

The results regarding the interaction between the HS treatments and the wheat varieties for TGW, BY and TGY are displayed in Table 7. The ANOVA showed that the HS treatment × wheat variety interaction did not have a significant effect on all the studied parameters. However, $V_3 \times HS_4$ and $V_3 \times HS_5$ produced the maximum values (40.46 and 46.41 g) for TGW, the $V_2 \times HS_5$ and $V_3 \times HS_4$ treatments had the greatest values for BY (5.11 and 4.05 kg plant$^{-1}$), and $V_3 \times HS_5$ and $V_1 \times HS_1$ were the best for TGY (7.079 and 8.152 ton ha$^{-1}$) in the 2018/2019 and 2019/2020 seasons, respectively. Based on the maximum and minimum obtained values in Table 7, the rates of increase were 12.39 and 32.75%, 121.10 and 76.09%, and 93.84 and 96.33% for TGW, BY and TGY in the 2018/2019 and 2019/2020 seasons, respectively.

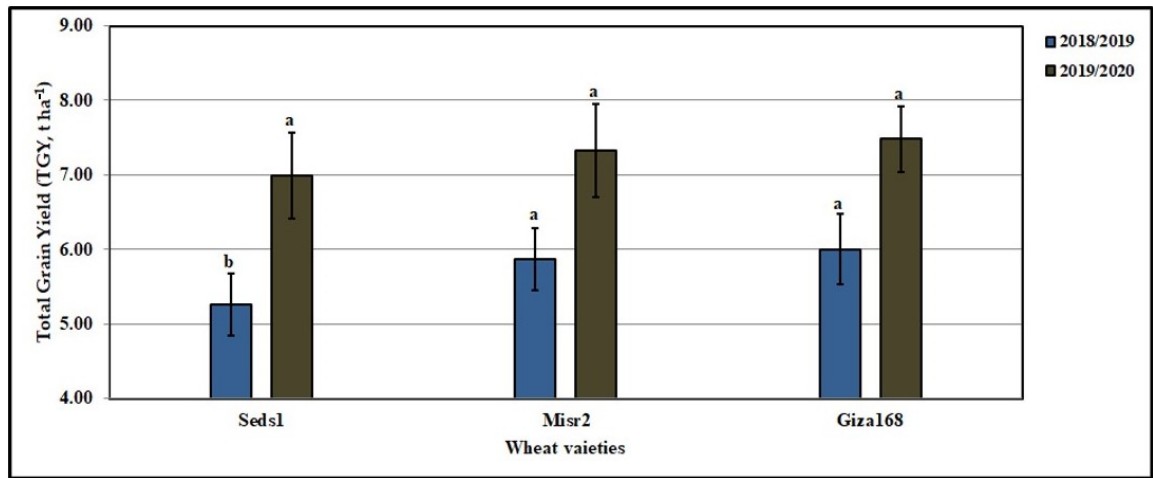

**Figure 18.** Influence of wheat varieties on total grain yield (TGY, t ha$^{-1}$) in clay loam soil in 2018/2019 and 2019/2020 seasons. Bars in the same years with a different letter indicate significant differences between treatments at $p \leq 0.01$. HS$_1$, HS$_2$, HS$_3$ represent HS applied as foliar spray at 1.0, 2.0, 4.0 g L$^{-1}$, respectively, HS$_4$, HS$_5$, HS$_6$ represent HS applied as soil application at 5.04, 7.56, 10.08 kg ha$^{-1}$.

*3.3. Influence of Humic Substances, Wheat Varieties and Their Interactions on Leaf Nitrogen, Phosphorus and Potassium Content*

3.3.1. Influence of Humic Substances

The results depicted in Figures 19–21 show the influence of different HS treatments on wheat leaf nitrogen (N), phosphorus (P) and potassium (K) content during the 2018/2019 and 2019/2020 seasons. The results indicated that the HS$_0$ and HS$_1$ treatments were the best treatments for leaf N contents. These treatments produced the maximum leaf N content (3.62 ad 4.06%). The HS$_1$ treatment gave the highest leaf P contents (2.22 and 1.98%) in 2018–2019 and 2019–2020, respectively. In contrast, the HS$_0$ treatment had the maximum content of K (3.32 and 3.36%) in the first and second seasons, respectively. Generally, the highest leaf N contents in the first season and K in both seasons was produced by untreated plants.

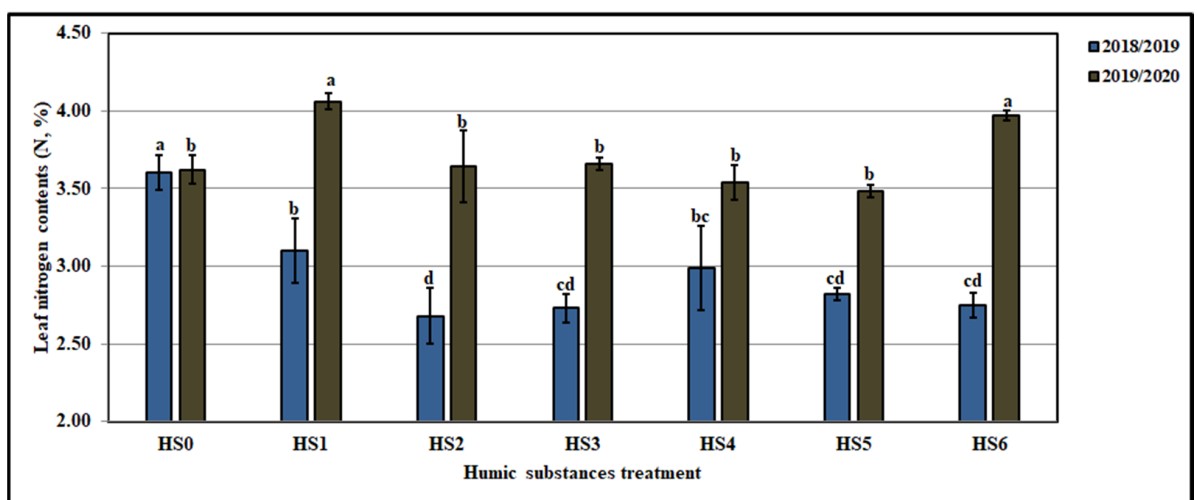

**Figure 19.** Influence of humic substances treatments on leaf nitrogen contents (N, %) of some wheat varieties grown in clay loam soil in 2018/2019 and 2019/2020 seasons. Bars in the same years with a different letter indicate significant differences between treatments at $p \leq 0.01$. HS$_1$, HS$_2$, HS$_3$ represent HS applied as foliar spray at 1.0, 2.0, 4.0 g L$^{-1}$, respectively, HS$_4$, HS$_5$, HS$_6$ represent HS applied as soil application at 5.04, 7.56, 10.08 kg ha$^{-1}$.

**Table 7.** Influence of humic substances doses on 1000-grain weight, biological yield and total grain yield of some wheat varieties grown in clay soil in 2018/2019 and 2019/2020 seasons.

| Treatment | (g) | | | (kg Plant$^{-1}$) | | | (t ha$^{-1}$) | | |
| | TGW | | | BY | | | TGY | | |
| HS | $V_1$ | $V_2$ | $V_3$ | $V_1$ | $V_2$ | $V_3$ | $V_1$ | $V_2$ | $V_3$ |
|---|---|---|---|---|---|---|---|---|---|
| **2018/2019 season** | | | | | | | | | |
| $HS_0$ | 36.57 $^a$ ± 1.11 | 38.54 $^{d–g}$ ± 1.19 | 46.33 $^{a–e}$ ± 1.15 | 2.92 $^{de}$ ± 0.21 | 2.31 $^e$ ± 0.12 | 3.56 $^{cd}$ ± 0.12 | 3.84 h ± 0.16 | 3.75 $^h$ ± 0.20 | 3.67 $^h$ ± 0.06 |
| $HS_1$ | 40.00 $^{c–g}$ ± 3.06 | 38.00 $^{e–g}$ ± 2.31 | 49.02 $^{ab}$ ± 2.27 | 4.76 $^{ab}$ ± 0.33 | 4.88 $^{ab}$ ± 0.45 | 4.15 $^{a–c}$ ± 0.41 | 6.56 $^{a–e}$ ± 0.29 | 6.90 $^{ab}$ ± 0.33 | 6.36 $^{a–e}$ ± 0.74 |
| $HS_2$ | 39.33 $^{d–g}$ ± 2.67 | 42.00 $^{b–g}$ ± 2.00 | 47.76 $^{a–c}$ ± 2.45 | 4.20 $^{a–c}$ ± 0.14 | 4.69 $^{ab}$ ± 0.51 | 4.25 $^{a–c}$ ± 0.26 | 5.22 $^{e–g}$ ± 1.00 | 6.54 $^{a–d}$ ± 0.72 | 6.54 $^{a–e}$ ± 0.51 |
| $HS_3$ | 42.19 $^{b–g}$ ± 0.99 | 39.67 $^{c–g}$ ± 3.93 | 47.74 $^{a–c}$ ± 2.71 | 3.86 $^{bc}$ ± 0.45 | 4.83 $^{ab}$ ± 0.16 | 4.58 $^{a–c}$ ± 0.02 | 4.72 $^{f–h}$ ± 0.41 | 5.32 $^{d–g}$ ± 0.88 | 6.52 $^{a–e}$ ± 0.23 |
| $HS_4$ | 42.67 $^{a–g}$ ± 1.76 | 38.67 $^{d–g}$ ± 5.33 | 50.46 $^a$ ± 0.77 | 4.00 $^{bc}$ ± 0.24 | 4.76 $^{ab}$ ± 0.61 | 4.37 $^{a–c}$ ± 0.44 | 5.51 $^{b–g}$ ± 0.38 | 6.85 $^{a–c}$ ± 0.45 | 6.05 $^{a–f}$ ± 0.70 |
| $HS_5$ | 41.33 $^{b–g}$ ± 1.76 | 44.67 $^{a–f}$ ± 1.76 | 46.86 $^{a–d}$ ± 2.39 | 4.87 $^{ab}$ ± 0.63 | 5.11 $^a$ ± 0.68 | 4.77 $^{ab}$ ± 0.15 | 6.47 $^{a–e}$ ± 0.22 | 6.07 $^{a–f}$ ± 0.13 | 7.08 $^a$ ± 0.41 |
| $HS_6$ | 36.00 $^g$ ± 3.06 | 40.67 $^{c–g}$ ± 1.76 | 44.41 $^{a–f}$ ± 5.30 | 4.09 $^{a–c}$ ± 0.21 | 4.43 $^{a–c}$ ± 0.19 | 4.42 $^{a–c}$ ± 0.32 | 4.46 $^{gh}$ ± 0.46 | 5.42 $^{c–g}$ ± 0.14 | 5.80 $^{a–g}$ ± 0.66 |
| **2019/2020 season** | | | | | | | | | |
| $HS_0$ | 36.76 $^{fg}$ ± 1.73 | 38.52 $^{e–g}$ ± 2.89 | 46.28 $^a$ ± 1.73 | 2.90 $^{bc}$ ± 0.12 | 2.27 $^c$ ± 0.18 | 3.52 $^{ab}$ ± 0.15 | 4.921 $^a$ ± 0.36 | 4.292 $^a$ ± 0.09 | 4.396 $^a$ ± 0.07 |
| $HS_1$ | 39.77 $^{c–g}$ ± 0.75 | 39.33 $^{d–g}$ ± 0.51 | 45.30 $^{a–e}$ ± 0.52 | 3.55 $^{ab}$ ± 0.33 | 3.58 $^{ab}$ ± 0.25 | 3.92 $^a$ ± 0.10 | 8.152 $^a$ ± 1.01 | 7.363 $^a$ ± 0.27 | 7.188 $^a$ ± 1.24 |
| $HS_2$ | 39.41 $^{d–g}$ ± 0.42 | 39.23 $^{d–g}$ ± 0.49 | 45.83 $^{ab}$ ± 0.39 | 3.23 $^{ab}$ ± 0.09 | 3.67 $^{ab}$ ± 0.40 | 3.82 $^a$ ± 0.06 | 7.346 $^a$ ± 0.31 | 6.996 $^a$ ± 0.65 | 7.914 $^a$ ± 0.09 |
| $HS_3$ | 40.38 $^{b–g}$ ± 0.90 | 34.96 $^g$ ± 3.47 | 44.66 $^{a–d}$ ± 2.44 | 3.44 $^{ab}$ ± 0.09 | 3.43 $^{ab}$ ± 0.24 | 3.67 $^{ab}$ ± 0.42 | 6.517 $^a$ ± 0.88 | 7.664 $^a$ ± 0.52 | 7.339 $^a$ ± 0.49 |
| $HS_4$ | 41.17 $^{a–f}$ ± 1.27 | 43.57 $^{a–e}$ ± 5.31 | 45.22 $^{a–c}$ ± 0.92 | 3.32 $^{ab}$ ± 0.09 | 3.41 $^{ab}$ ± 0.08 | 4.05 $^a$ ± 0.12 | 7.576 $^a$ ± 0.13 | 8.022 $^a$ ± 1.04 | 7.910 $^a$ ± 0.68 |
| $HS_5$ | 36.40 $^{fg}$ ± 2.68 | 38.50 $^{e–g}$ ± 0.17 | 46.41 $^a$ ± 0.38 | 3.24 $^{ab}$ ± 0.33 | 3.34 $^{ab}$ ± 0.02 | 3.55 $^{ab}$ ± 0.07 | 6.153 $^a$ ± 0.88 | 7.426 $^a$ ± 0.49 | 8.109 $^a$ ± 0.25 |
| $HS_6$ | 38.90 $^{e–g}$ ± 0.57 | 38.00 $^{e–g}$ ± 1.64 | 45.24 $^{a–c}$ ± 0.63 | 3.27 $^{ab}$ ± 0.35 | 3.32 $^{ab}$ ± 0.59 | 3.91 $^a$ ± 0.17 | 6.391 $^a$ ± 0.44 | 7.544 $^a$ ± 1.30 | 7.480 $^a$ ± 0.28 |

Means in the same column denoted by a different letter indicate significant difference between treatments at ≤0.05. $HS_1$, $HS_2$, $HS_3$ represent HS applied as foliar spray at 1.0, 2.0, 4.0 g L$^{-1}$, respectively, $HS_4$, $HS_5$, $HS_6$ represent HS applied as soil application at 5.04, 7.56, 10.08 kg ha$^{-1}$. $V_1$ = Seds1 cv, $V_2$ = Misr2 cv, $V_3$ = Giza168 cv. TGW = 1000-grain weight, BY = Biological yield, TGY = Total grain yield.

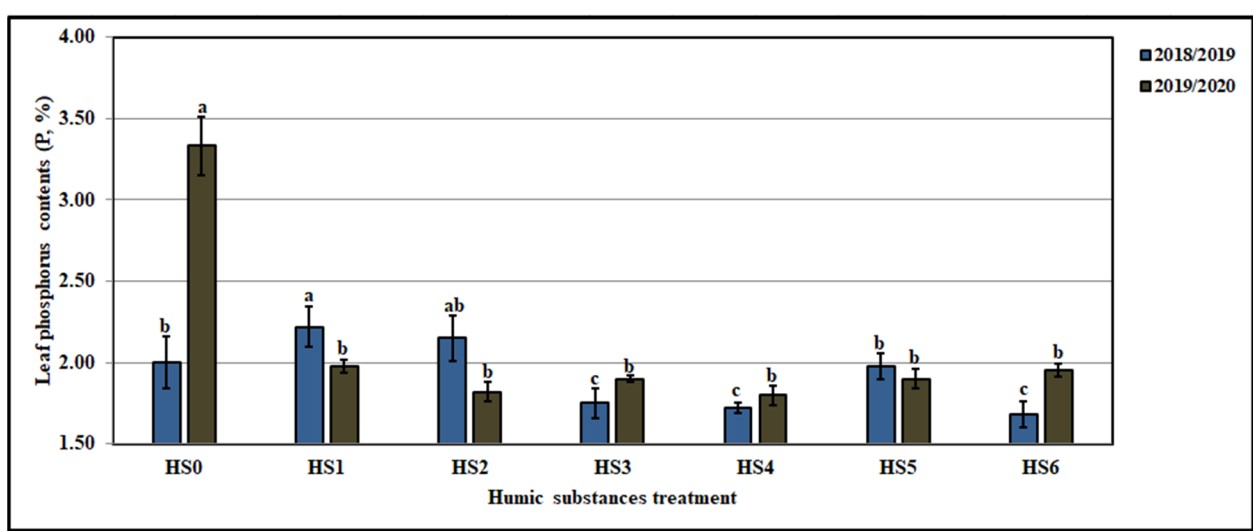

**Figure 20.** Influence of humic substances treatments on leaf phosphorus contents (P, %) of some wheat varieties grown in clay loam soil in 2018/2019 and 2019/2020 seasons. Bars in the same years with a different letter indicate significant differences between treatments at $p \leq 0.01$. $HS_1$, $HS_2$, $HS_3$ represent HS applied as foliar spray at 1.0, 2.0, 4.0 g $L^{-1}$, respectively, $HS_4$, $HS_5$, $HS_6$ represent HS applied as soil application at 5.04, 7.56, 10.08 kg $ha^{-1}$.

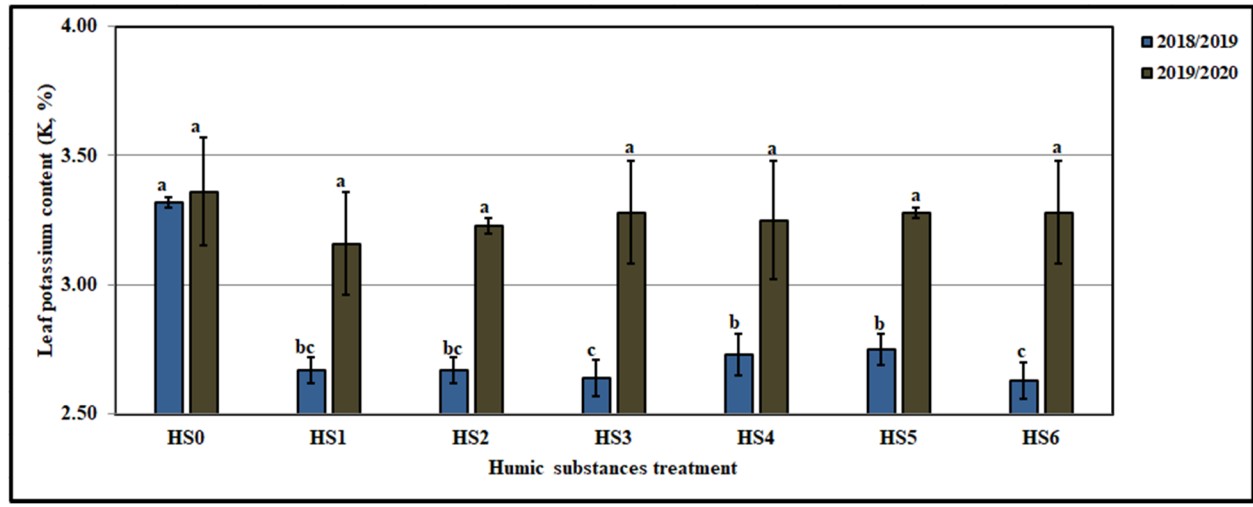

**Figure 21.** Influence of humic substances treatments on leaf potassium contents (K, %) of some wheat varieties grown in clay loam soil in 2018/2019 and 2019/2020 seasons. Bars in the same years with a different letter indicate significant differences between treatments at $p \leq 0.01$. $HS_1$, $HS_2$, $HS_3$ represent HS applied as foliar spray at 1.0, 2.0, 4.0 g $L^{-1}$, respectively, $HS_4$, $HS_5$, $HS_6$ represent HS applied as soil application at 5.04, 7.56, 10.08 kg $ha^{-1}$.

### 3.3.2. Influence of Wheat Varieties

The results of the ANOVA indicated that the variety had no significant effect on leaf P contents in the first season and on K content in the second season only, but had a significant influence (at $p \leq 0.01$) on the leaf N and K contents in the first season and a significant influence (at $p \leq 0.05$) on leaf N contents in the second season and P content in the second season. As shown in Figures 22–24, different data were obtained regarding the influence of wheat varieties on leaf N, P and K contents. However, $V_3$ and $V_1$ plants were the superior variety for leaf N content (3.13 and 3.80%) in the first and second seasons, respectively, while $V_1$ plants produced the highest values for leaf P contents (1.98 and 2.02%) in both

seasons. On the other hand, $V_2$ and $V_3$ plants were the best for leaf K contents (2.87 and 3.29%). The data indicated that the rate of increase reached 13.82 and 4.40%, 4.76 and 16.09%, and 6.50 and 1.55% for leaf N, P and K contents in the 2018/2019 and 2019/2020 seasons, respectively.

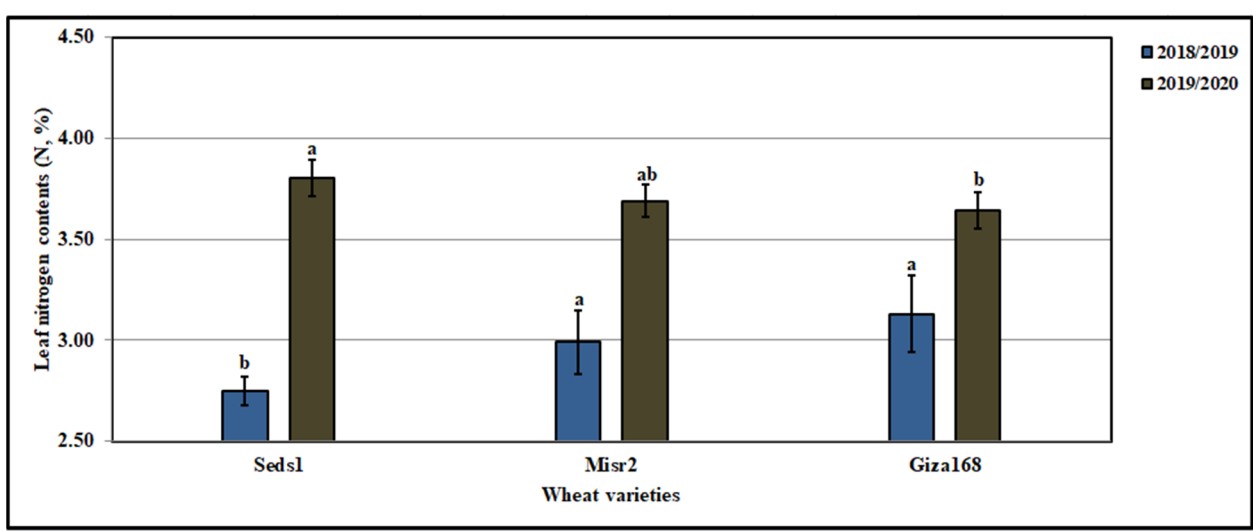

**Figure 22.** Influence of wheat varieties on leaf nitrogen contents (N, %) in clay loam soil in 2018/2019 and 2019/2020 seasons. Bars in the same years with a different letter indicate significant differences between treatments at $p \leq 0.01$. $HS_1$, $HS_2$, $HS_3$ represent HS applied as foliar spray at 1.0, 2.0, 4.0 g $L^{-1}$, respectively, $HS_4$, $HS_5$, $HS_6$ represent HS applied as soil application at 5.04, 7.56, 10.08 kg $ha^{-1}$.

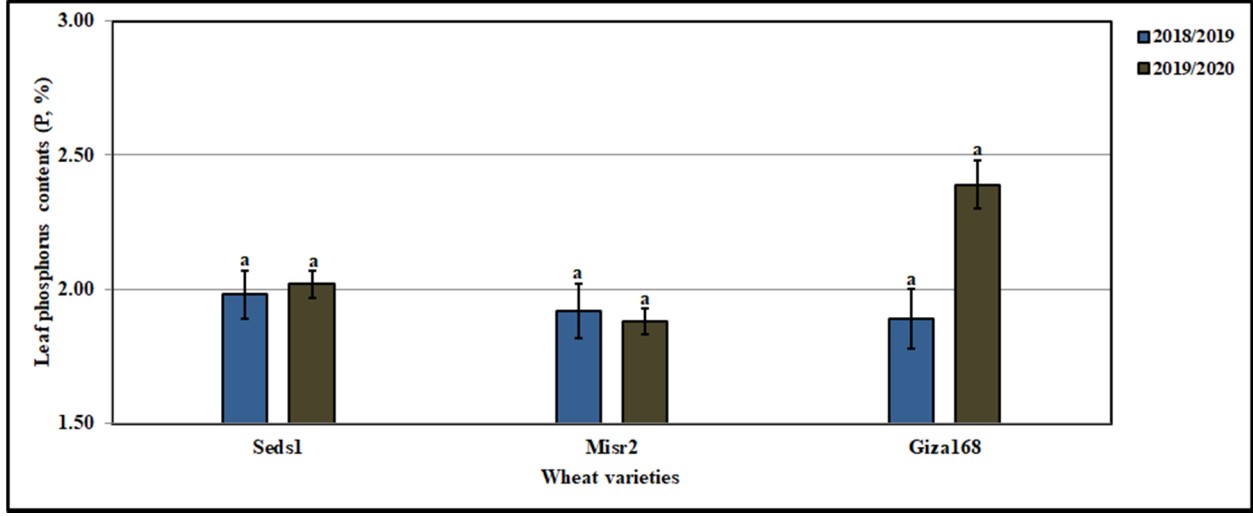

**Figure 23.** Influence of wheat varieties on leaf phosphorus contents (P, %) in clay loam soil in 2018/2019 and 2019/2020 seasons. Bars in the same years with a different letter indicate significant differences between treatments at $p \leq 0.01$. $HS_1$, $HS_2$, $HS_3$ represent HS applied as foliar spray at 1.0, 2.0, 4.0 g $L^{-1}$, respectively, $HS_4$, $HS_5$, $HS_6$ represent HS applied as soil application at 5.04, 7.56, 10.08 kg $ha^{-1}$.

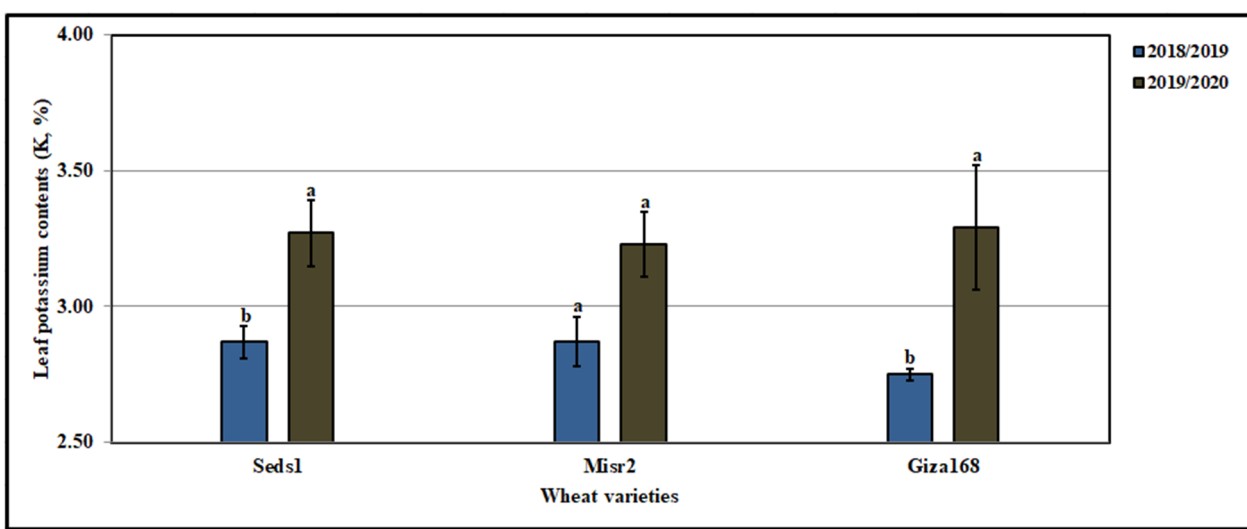

**Figure 24.** Influence of wheat varieties on leaf potassium contents (K, %) in clay loam soil in 2018/2019 and 2019/2020 seasons. Bars in the same years with a different letter indicate significant differences between treatments at $p \leq 0.01$. $HS_1$, $HS_2$, $HS_3$ represent HS applied as foliar spray at 1.0, 2.0, 4.0 g $L^{-1}$, respectively, $HS_4$, $HS_5$, $HS_6$ represent HS applied as soil application at 5.04, 7.56, 10.08 kg $ha^{-1}$.

### 3.3.3. Influence of the Humic Substance × Wheat Variety Interaction

The results in Table 8 show the effect of the interaction between HS and wheat variety on wheat leaf N, P and K contents. Analysis of variance showed that all varieties had a significant influence (at $p \leq 0.01$) on the leaf N content in the 2018/2019 season and leaf K content in the 2019/2020 season, as well as leaf P contents in both seasons. The results revealed that the best values (3.71 and 4.27) for N content were obtained by untreated Giza168 ($V_2 \times HS_0$) and Seds1 plants nourished with 4.0 g $L^{-1}$ of HS ($V_1 \times HS_3$). Moreover, $HS_1 \times V_3$ and $HS_3 \times V_3$ were the superior treatments for leaf P content, and $HS_0 \times V_2$ and $HS_0 \times V_1$ were the best for K content in both seasons, respectively. The results in Table 8 indicated that the contents of N increased by 47.22 and 42.33%, P content increased by 70.83 and 315.82%, and K increased by 40.66 and 120.88% in the first and second seasons, respectively.

### 3.4. Influence of Humic Substances, Wheat Varieties and Their Interactions on Micronutrient Contents

### 3.4.1. Influence of Humic Substances

The influence of HS on the leaf content of micronutrients such as iron (Fe), manganese (Mn), zinc (Zn) and copper (Cu) in the 2018/2019 and 2019/2020 seasons are shown in Figures 25–28. The maximum values of nutrient content (331.00, 30.42, 7.70 and 4.53 mg $kg^{-1}$) were obtained for $HS_3$, $HS_6$, $HS_5$, $HS_6$ treatments in the first season, respectively, and the maximum values (700.7, 32.47, 14.60 and 6.19 mg $kg^{-1}$ for Fe, Mn, Zn and Cu) were produced by the $HS_2$ treatment in the second season. The general trend of the data portrayed in Figures 25–28 indicated that the nutrient leaf contents doubled in the second season compared with the first season. These results could be due to soil content of those nutrients being higher than in first season, as mentioned in Table 4, as well as the residual impact of HS treatments in the first season. The results of the ANOVA indicate that all HS treatments had a significant influence (at $p \leq 0.01$) on the leaf Fe, Mn, Zn and Cu contents in both seasons. The obtained data indicated that the rate of increase reached 45.88 and 18.70% for Fe, 48.83 and 33.83% for Mn, 83.33 and 60.97% for Zn, and 87.19 and 57.91% for Cu in the first and second seasons, respectively.

**Table 8.** Influence of humic substances doses on some leaf nitrogen, phosphorus and potassium contents of some wheat varieties grown in clay soil in 2018/2019 and 2019/2020 seasons.

| Treatment | (%) | | | | | | | | |
|---|---|---|---|---|---|---|---|---|---|
| | N | | | P | | | K | | |
| HS | $V_1$ | $V_2$ | $V_3$ | $V_1$ | $V_2$ | $V_3$ | $V_1$ | $V_2$ | $V_3$ |
| **2018/2019 season** | | | | | | | | | |
| $HS_0$ | 2.73 [cd] ± 0.08 | 2.94 [cd] ± 0.08 | 3.64 [a] ± 0.16 | 1.95 [b–g] ± 0.18 | 1.95 [b–g] ± 0.19 | 2.08 [a–e] ± 0.10 | 3.37 [a] ± 0.02 | 3.39 [a] ± 0.01 | 3.19 [b] ± 0.01 |
| $HS_1$ | 3.50 [e–g] ± 0.12 | 3.64 [d–g] ± 0.40 | 3.71 [d–f] ± 0.12 | 2.17 [a–d] ± 0.07 | 2.04 [b–f] ± 0.18 | 2.46 [a] ± 0.10 | 2.58 [f–h] ± 0.09 | 2.82 [cd] ± 0.05 | 2.60 [e–h] ± 0.01 |
| $HS_2$ | 2.52 [d] ± 0.08 | 2.73 [cd] ± 0.12 | 2.80 [cd] ± 0.32 | 2.21 [a–c] ± 0.08 | 2.19 [a–c] ± 0.12 | 2.06 [b–e] ± 0.22 | 2.70 [d–g] ± 0.02 | 2.74 [c–f] ± 0.12 | 2.56 [gh] ± 0.01 |
| $HS_3$ | 2.52 [d] ± 0.08 | 2.80 [cd] ± 0.16 | 2.87 [cd] ± 0.04 | 1.99 [b–g] ± 0.14 | 1.80 [c–h] ± 0.03 | 1.44 [h] ± 0.12 | 2.52 [hi] ± 0.01 | 2.70 [d–g] ± 0.17 | 2.68 [d–h] ± 0.01 |
| $HS_4$ | 2.73 [cd] ± 0.04 | 3.15 [bc] ± 0.20 | 3.08 [bc] ± 0.58 | 1.73 [e–h] ± 0.01 | 1.79 [dh] ± 0.03 | 1.65 [f–h] ± 0.06 | 2.62 [e–h] ± 0.02 | 2.82 [cd] ± 0.17 | 2.76 [c–e] ± 0.03 |
| $HS_5$ | 2.73 [cd] ± 0.04 | 2.87 [cd] ± 0.04 | 2.87 [cd] ± 0.04 | 2.22 [ab] ± 0.03 | 1.63 [gh] ± 0.10 | 2.08 [a–e] ± 0.13 | 2.68 [d–h] ± 0.03 | 2.88 [c] ± 0.10 | 2.70 [d–g] ± 0.05 |
| $HS_6$ | 2.52 [d] ± 0.05 | 2.87 [cd] ± 0.11 | 2.94 [cd] ± 0.08 | 1.55 [h] ± 0.12 | 2.04 [b–f] ± 0.08 | 1.45 [h] ± 0.03 | 2.41 [i] ± 0.19 | 2.76 [c–e] ± 0.01 | 2.72 [c–g] ± 0.01 |
| **2019/2020 season** | | | | | | | | | |
| $HS_0$ | 3.50 [e–g] ± 0.08 | 3.64 [d–g] ± 0.08 | 3.71 [d–f] ± 0.12 | 1.77 [b] ± 0.01 | 1.66 [b] ± 0.07 | 1.88 [a] ± 0.47 | 3.49 [a] ± 0.58 | 3.37 [bc] ± 0.05 | 3.23 [f–h] ± 0.01 |
| $HS_1$ | 4.20 [a] ± 0.08 | 4.13 [ab] ± 0.04 | 3.85 [b–d] ± 0.04 | 2.06 [b] ± 0.02 | 2.06 [b] ± 0.06 | 1.76 [b] ± 0.03 | 3.12 [l] ± 0.01 | 3.15 [j–l] ± 0.01 | 3.21 [g–i] ± 0.58 |
| $HS_2$ | 3.64 [d–g] ± 0.24 | 3.50 [e–g] ± 0.29 | 3.78 [c–e] ± 0.16 | 1.88 [b] ± 0.06 | 1.88 [b] ± 0.08 | 1.71 [b] ± 0.05 | 3.13 [kl] ± 0.08 | 3.27 [ef] ± 0.01 | 3.27 [ef] ± 0.01 |
| $HS_3$ | 4.27 [a] ± 0.04 | 3.00 [h] ± 0.05 | 3.71 [d–f] ± 0.04 | 1.92 [b] ± 0.02 | 1.92 [b] ± 0.02 | 1.63 [b] ± 0.01 | 3.31 [de] ± 0.01 | 3.21 [g–i] ± 0.00 | 3.33 [cd] ± 0.59 |
| $HS_4$ | 3.78 [c–e] ± 0.08 | 3.85 [b–d] ± 0.04 | 3.00 [h] ± 0.21 | 1.80 [b] ± 0.07 | 1.80 [b] ± 0.06 | 1.67 [b] ± 0.04 | 3.21 [g–i] ± 0.12 | 3.25 [fg] ± 0.18 | 3.27 [ef] ± 0.40 |
| $HS_5$ | 3.43 [fg] ± 0.05 | 3.64 [d–g] ± 0.04 | 3.36 [g] ± 0.03 | 2.49 [b] ± 0.16 | 2.49 [b] ± 0.01 | 1.76 [b] ± 0.02 | 3.37 [bc] ± 0.02 | 3.19 [gi] ± 0.01 | 3.27 [ef] ± 0.01 |
| $HS_6$ | 3.78 [c–e] ± 0.03 | 4.06 [a–c] ± 0.03 | 4.06 [a–c] ± 0.02 | 2.21 [b] ± 0.03 | 2.21 [b] ± 0.04 | 1.76 [b] ± 0.04 | 3.27 [ef] ± 0.01 | 3.17 [i–k] ± 0.58 | 3.41 [b] ± 0.02 |

Means in the same column denoted by a different letter indicate significant difference between treatments at ≤0.05. $HS_1$, $HS_2$, $HS_3$ represent HS applied as foliar spray at 1.0, 2.0, 4.0 g $L^{-1}$, respectively, $HS_4$, $HS_5$, $HS_6$ represent HS applied as soil application at 5.04, 7.56, 10.08 kg $ha^{-1}$. $V_1$ = Seds1 cv, $V_2$ = Misr2 cv, $V_3$ = Giza168 cv, N = nitrogen content, P = phosphorus content and K = potassium content.

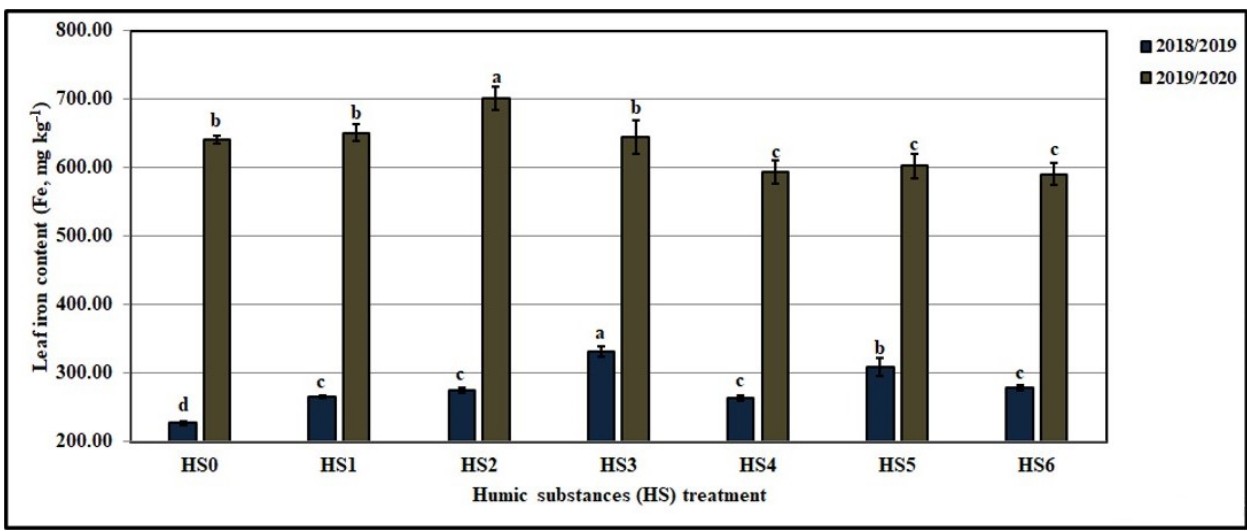

**Figure 25.** Influence of humic substances treatments on leaf iron contents (Fe, mg kg$^{-1}$) of some wheat varieties grown in clay loam soil in 2018/2019 and 2019/2020 seasons. Bars in the same years with a different letter indicate significant differences between treatments at $p \leq 0.01$. HS$_1$, HS$_2$, HS$_3$ represent HS applied as foliar spray at 1.0, 2.0, 4.0 g L$^{-1}$, respectively, HS$_4$, HS$_5$, HS$_6$ represent HS applied as soil application at 5.04, 7.56, 10.08 kg ha$^{-1}$.

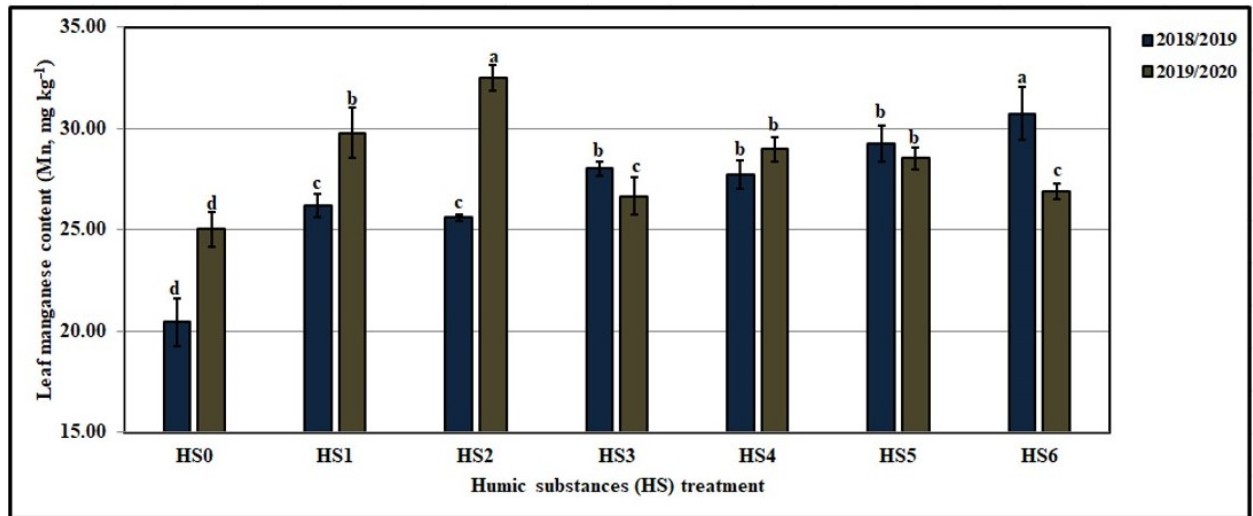

**Figure 26.** Influence of humic substances treatments on leaf manganese contents (Mn, mg kg$^{-1}$) of some wheat varieties grown in clay loam soil in 2018/2019 and 2019/2020 seasons. Bars in the same years with a different letter indicate significant differences between treatments at $p \leq 0.01$. HS$_1$, HS$_2$, HS$_3$ represent HS applied as foliar spray at 1.0, 2.0, 4.0 g L$^{-1}$, respectively, HS$_4$, HS$_5$, HS$_6$ represent HS applied as soil application at 5.04, 7.56, 10.08 kg ha$^{-1}$.

### 3.4.2. Influence of Wheat Varieties

The effect of variety on all the above-mentioned nutrient contents of wheat plants in both seasons is shown in Figures 29–32. The results clearly indicated that the varieties, in descending order, were ranked as $V_1 > V_3 > V_2$ and $V_3 > V_2 > V_1$ for Fe, $V_2 > V_3 > V_1$ and $V_3 > V_1 > V_2$ for Mn, $V_3 > V_2 > V_1$ and $V_3 > V_2 > V_1$ for Zn, and $V_3 > V_1 > V_2$ and $V_3 > V_2 > V_1$ for Cu in the first and second seasons, respectively. However, the highest values were 321.40 and 665.10 for Fe, 29.54 and 30.52 for Mn, 7.58 and 13.09 for Zn, and 3.69 and 6.46 for Cu, whereas the lowest values were 242.70 and 597.80 for Fe, 23.46 and

26.40 for Mn, 4.09 and 9.09 for Zn, and 3.14 and 3.89 mg kg$^{-1}$ for Cu in the 2018/2019 and 2019/2020 seasons, respectively. It was noticed that the percentages of increase between the highest and lowest values were 32.43 and 11.26% for Fe, 25.92 and 15.91% for Mn, 85.33 and 44.00% for Zn, and 17.52 and 66.07% for Cu in the first and second seasons, respectively. The analysis of variance revealed that all the above-mentioned nutrient contents were highly significantly affected by variety.

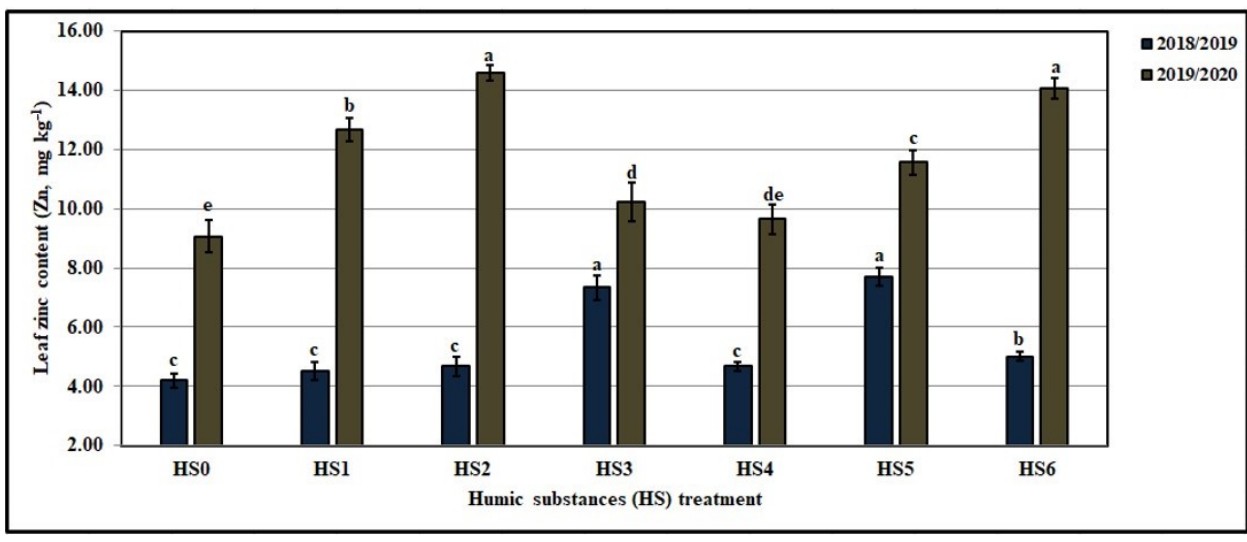

**Figure 27.** Influence of humic substances treatments on leaf zinc contents (Zn, mg kg$^{-1}$) of some wheat varieties grown in clay loam soil in 2018/2019 and 2019/2020 seasons. Bars in the same years with a different letter indicate significant differences between treatments at $p \leq 0.01$. HS$_1$, HS$_2$, HS$_3$ represent HS applied as foliar spray at 1.0, 2.0, 4.0 g L$^{-1}$, respectively, HS$_4$, HS$_5$, HS$_6$ represent HS applied as soil application at 5.04, 7.56, 10.08 kg ha$^{-1}$.

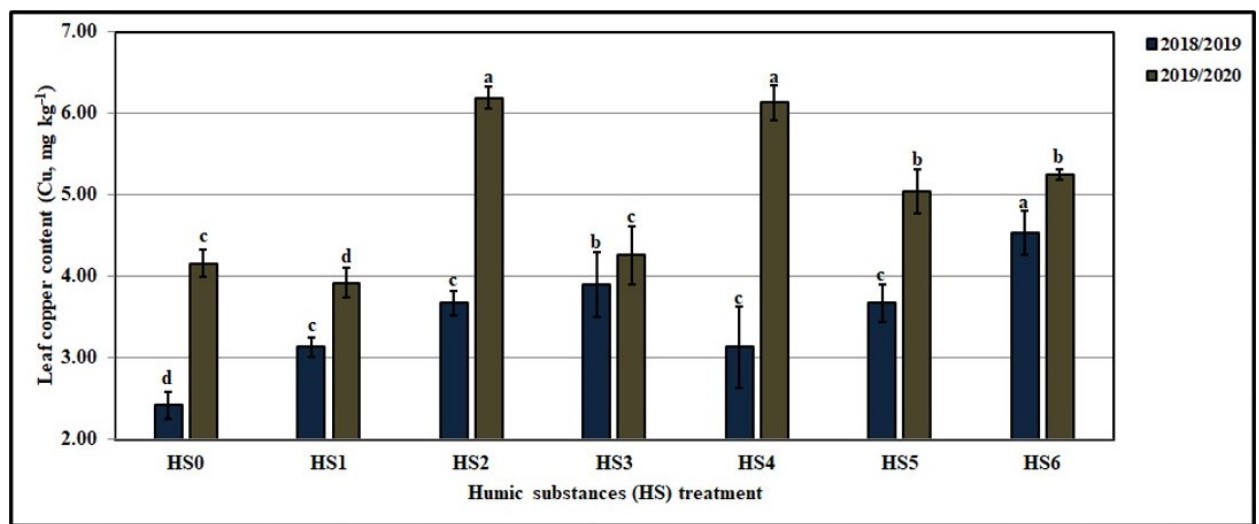

**Figure 28.** Influence of humic substances treatments on leaf copper contents (Cu, mg kg$^{-1}$) of some wheat varieties grown in clay loam soil in 2018/2019 and 2019/2020 seasons. Bars in the same years with a different letter indicate significant differences between treatments at $p \leq 0.01$. HS$_1$, HS$_2$, HS$_3$ represent HS applied as foliar spray at 1.0, 2.0, 4.0 g L$^{-1}$, respectively, HS$_4$, HS$_5$, HS$_6$ represent HS applied as soil application at 5.04, 7.56, 10.08 kg ha$^{-1}$.

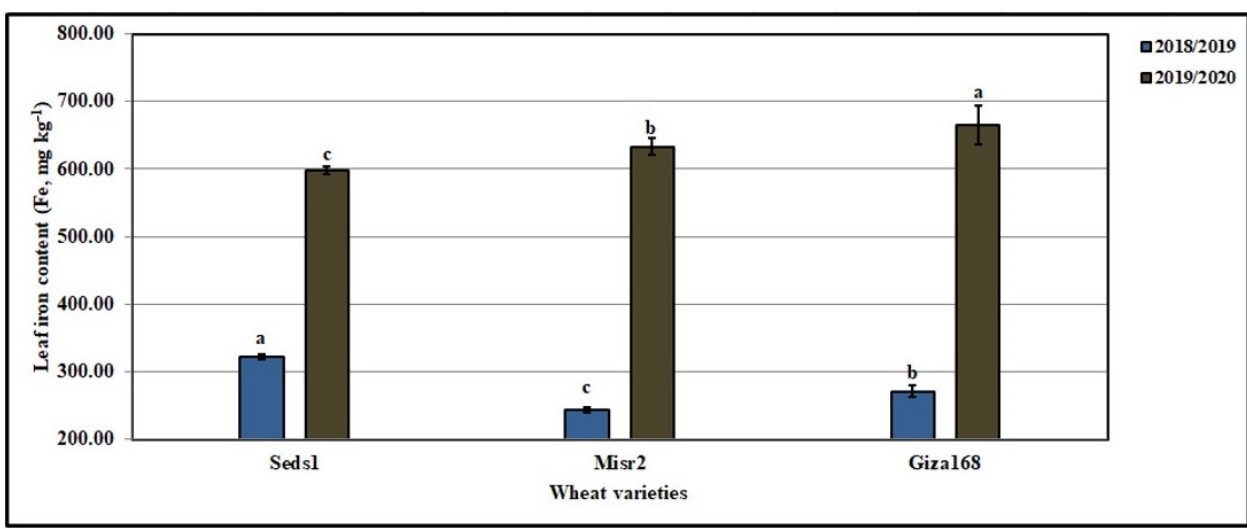

**Figure 29.** Influence of wheat varieties on leaf iron contents (Fe, mg kg$^{-1}$) in clay loam soil in 2018/2019 and 2019/2020 seasons. Bars in the same years with a different letter indicate significant differences between treatments at $p \leq 0.01$. HS$_1$, HS$_2$, HS$_3$ represent HS applied as foliar spray at 1.0, 2.0, 4.0 g L$^{-1}$, respectively, HS$_4$, HS$_5$, HS$_6$ represent HS applied as soil application at 5.04, 7.56, 10.08 kg ha$^{-1}$.

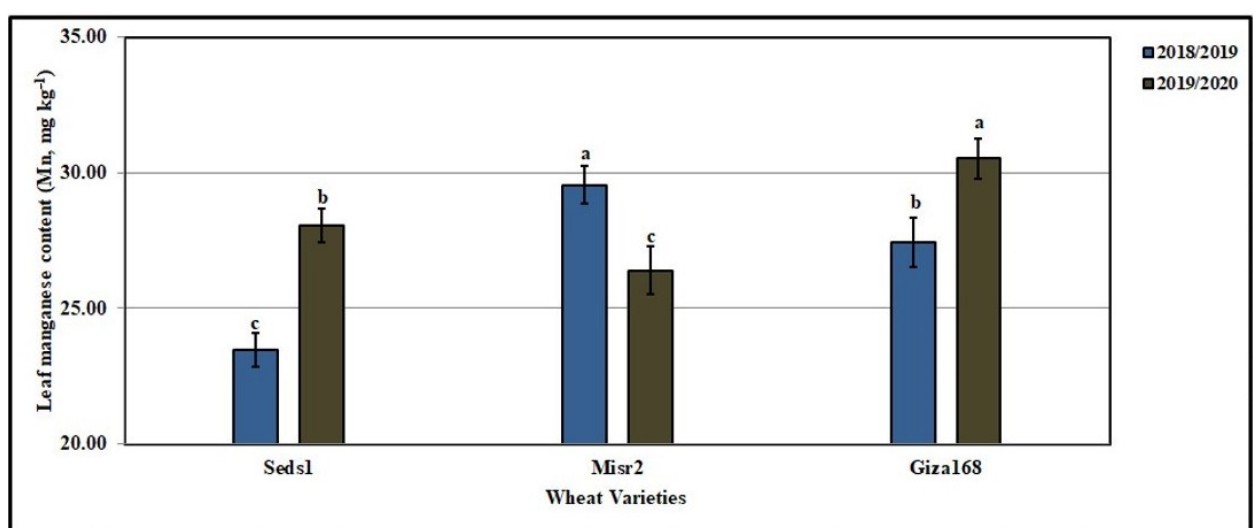

**Figure 30.** Influence of wheat varieties on leaf manganese contents (Mn, mg kg$^{-1}$) in clay loam soil in 2018/2019 and 2019/2020 seasons. Bars in the same years with a different letter indicate significant differences between treatments at $p \leq 0.01$. HS$_1$, HS$_2$, HS$_3$ represent HS applied as foliar spray at 1.0, 2.0, 4.0 g L$^{-1}$, respectively, HS$_4$, HS$_5$, HS$_6$ represent HS applied as soil application at 5.04, 7.56, 10.08 kg ha$^{-1}$.

### 3.4.3. Influence of the Humic Acid × Wheat Variety Interaction

The two-way interaction between HS and wheat variety showed highly significant effects on leaf nutrient contents in both growth seasons. It is clear from Table 9 that the HS$_5$ × V$_1$, HS$_6$ × V$_2$, HS$_5$ × V$_3$ and HS$_6$ × V$_3$ treatments produced the greatest values for leaf Fe, Mn, Zn and Cu contents (407.40, 35.40, 14.20 and 5.80 mg kg$^{-1}$ for Fe, Mn, Zn and Cu, respectively) in the 2018/2019 season, while the HS$_2$ × V$_3$ treatment was superior for all the above-mentioned nutrients, with maximum values of 822.03, 37.42, 19.81 and 9.40 mg kg$^{-1}$ for Fe, Mn, Zn and Cu, respectively, in the 2019/2020 season. The percentages of increase from the lowest value reached 90.37 and 69.43%, 93.13 and 102.38%, 235.70 and

219.52%, and 163.64 and 335.19% for Fe, Mn, Zn and Cu in the first and second seasons, respectively.

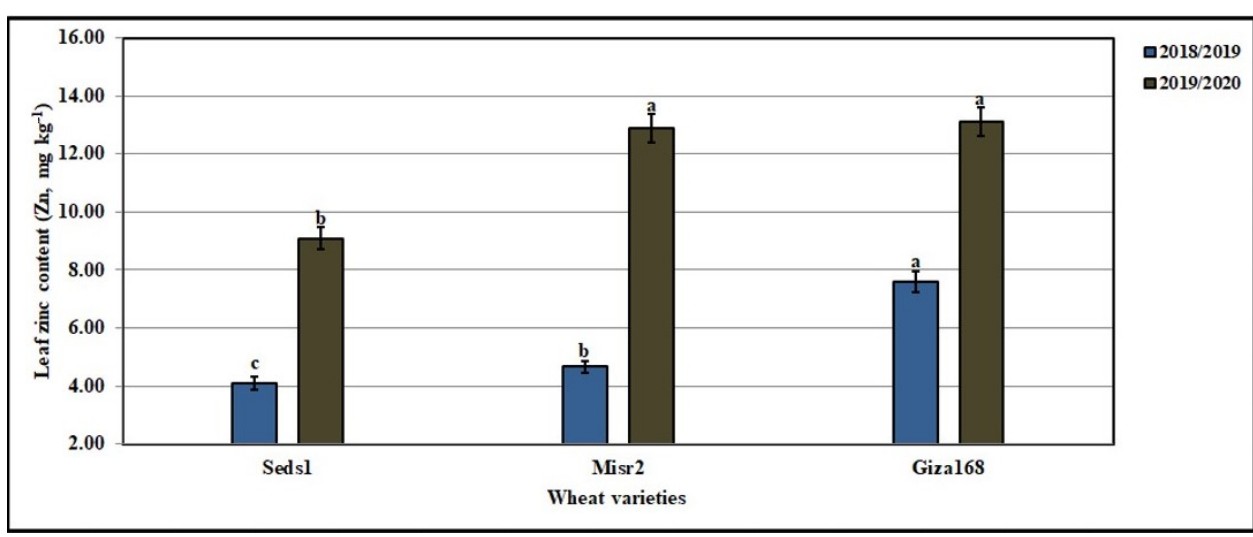

**Figure 31.** Influence of wheat varieties on leaf zinc contents (Zn, mg kg$^{-1}$) in clay loam soil in 2018/2019 and 2019/2020 seasons. Bars in the same years with a different letter indicate significant differences between treatments at $p \leq 0.01$. HS$_1$, HS$_2$, HS$_3$ represent HS applied as foliar spray at 1.0, 2.0, 4.0 g L$^{-1}$, respectively, HS$_4$, HS$_5$, HS$_6$ represent HS applied as soil application at 5.04, 7.56, 10.08 kg ha$^{-1}$.

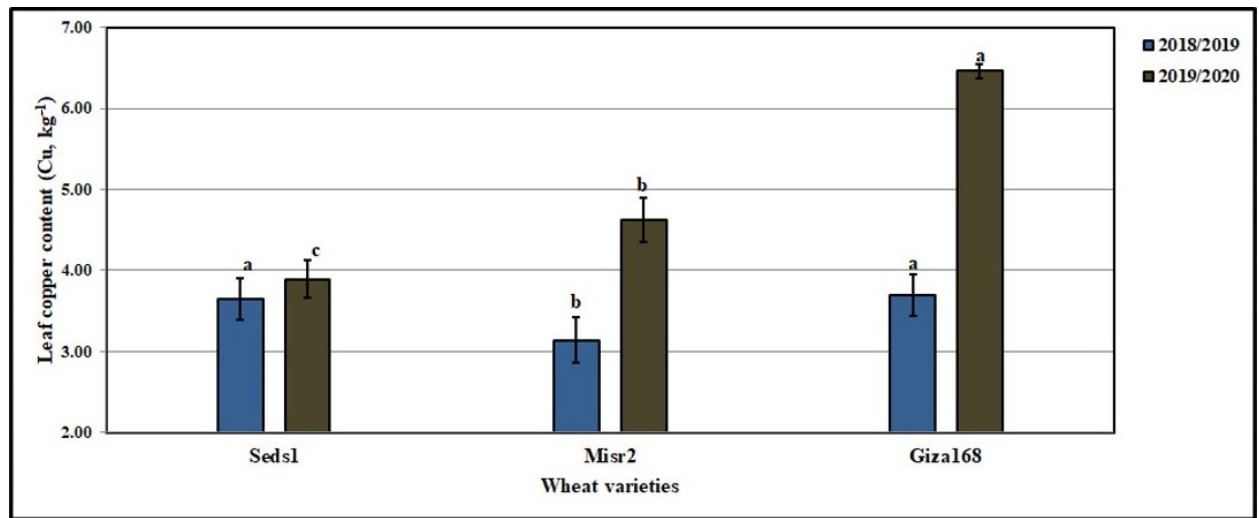

**Figure 32.** Influence of wheat varieties on leaf copper contents (Cu, mg kg$^{-1}$) in clay loam soil in 2018/2019 and 2019/2020 seasons. Bars in the same years with a different letter indicate significant differences between treatments at $p \leq 0.01$. HS$_1$, HS$_2$, HS$_3$ represent HS applied as foliar spray at 1.0, 2.0, 4.0 g L$^{-1}$, respectively, HS$_4$, HS$_5$, HS$_6$ represent HS applied as soil application at 5.04, 7.56, 10.08 kg ha$^{-1}$.

**Table 9.** Influence of humic substances doses on some leaf iron, manganese, zinc, and copper contents of some wheat varieties grown in clay soil in 2018/2019 and 2019/2020 seasons.

| Treatment | \multicolumn mg kg$^{-1}$ | | | | | | | | | | | |
|---|---|---|---|---|---|---|---|---|---|---|---|---|
| | Fe | | | Mn | | | Zn | | | Cu | | |
| HS | $V_1$ | $V_2$ | $V_3$ | $V_1$ | $V_2$ | $V_3$ | $V_1$ | $V_2$ | $V_3$ | $V_1$ | $V_2$ | $V_3$ |
| | \multicolumn 2018/2019 growth season | | | | | | | | | | | |
| HS$_0$ | 249.7 $^{g-i}$ ± 5.49 | 214.0 $^{l}$ ± 3.06 | 217.0 $^{j-l}$ ± 1.53 | 18.3 $^{m}$ ± 1.20 | 20.0 $^{lm}$ ± 1.15 | 23.00 $^{i-k}$ ± 1.15 | 3.5 $^{h-k}$ ± 0.15 | 4.2 $^{e-h}$ ± 0.19 | 4.9 $^{d-f}$ ± 0.34 | 2.8 $^{h-l}$ ± 0.30 | 2.3 $^{jl}$ ± 0.15 | 2.2 $^{l}$ ± 0.06 |
| HS$_1$ | 321.8 $^{cd}$ ± 1.96 | 217.0 $^{kl}$ ± 1.27 | 258.4 $^{f-h}$ ± 1.15 | 21.2 $^{kl}$ ± 0.23 | 30.4 $^{c-e}$ ± 0.46 | 27.00 $^{f-g}$ ± 1.04 | 3.6 $^{g-k}$ ± 0.58 | 5.0 $^{d-f}$ ± 0.23 | 5.0 $^{d-f}$ ± 0.12 | 4.4 $^{b-e}$ ± 0.12 | 2.8 $^{h-l}$ ± 0.12 | 2.2 $^{l}$ ± 0.12 |
| HS$_2$ | 338.8 $^{c}$ ± 5.31 | 251.2 $^{i-k}$ ± 5.89 | 234.0 $^{hl}$ ± 1.15 | 20.2 $^{lm}$ ± 0.12 | 29.6 $^{c-f}$ ± 0.23 | 27.00 $^{fg}$ ± 0.12 | 4.4 $^{e-h}$ ± 0.23 | 4.6 $^{d-g}$ ± 0.35 | 5.0 $^{d-f}$ ± 0.35 | 5.0 $^{ab}$ ± 0.12 | 3.4 $^{e-k}$ ± 0.23 | 2.6 $^{i-l}$ ± 0.12 |
| HS$_3$ | 338.9 $^{ab}$ ± 4.10 | 229.2 $^{i-l}$ ± 5.77 | 375.0 $^{b}$ ± 13.51 | 22.6 $^{j-l}$ ± 0.35 | 27.8 $^{e-g}$ ± 0.35 | 33.60 $^{ab}$ ± 0.46 | 4.0 $^{f-k}$ ± 0.23 | 5.2 $^{de}$ ± 0.23 | 12.8 $^{b}$ ± 0.81 | 3.2 $^{f-l}$ ± 0.58 | 3.9 $^{c-g}$ ± 0.06 | 4.60 $^{b-d}$ ± 0.58 |
| HS$_4$ | 244.4 $^{g-j}$ ± 3.70 | 262.0 $^{fg}$ ± 5.54 | 282.2 $^{ef}$ ± 2.42 | 24.6 ± 0.35 | 31.6 $^{b-d}$ ± 0.23 | 27.00 $^{fg}$ ± 1.50 | 5.6 $^{d}$ ± 0.23 | 4.2 $^{e-i}$ ± 0.12 | 4.2 $^{e-j}$ ± 0.12 | 3.0 $^{f-l}$ ± 0.35 | 2.8 $^{g-l}$ ± 0.58 | 3.60 $^{d-i}$ ± 0.58 |
| HS$_5$ | 407.4 $^{a}$ ± 3.58 | 233.0 $^{h-l}$ ± 2.42 | 287.7 $^{e}$ ± 32.44 | 30.4 $^{c-e}$ ± 0.69 | 32.0 $^{bc}$ ± 1.62 | 25.40 $^{g-i}$ ± 0.35 | 4.3 $^{e-h}$ ± 0.17 | 4.6 $^{d-g}$ ± 0.12 | 14.2 $^{a}$ ± 0.58 | 3.4 $^{e-j}$ ± 0.23 | 2.8 $^{h-l}$ ± 0.23 | 4.80 $^{bc}$ ± 0.23 |
| HS$_6$ | 298.8 $^{de}$ ± 2.54 | 292.2 $^{e}$ ± 2.42 | 242.6 $^{g-k}$ ± 5.89 | 26.9 $^{f-h}$ ± 1.33 | 35.4 $^{a}$ ± 0.81 | 29.00 $^{d-f}$ ± 1.73 | 3.2 $^{ik}$ ± 0.12 | 4.8 $^{d-f}$ ± 0.23 | 7.0 $^{c}$ ± 0.12 | 3.8 $^{c-h}$ ± 0.12 | 4.0 $^{b-f}$ ± 0.58 | 5.80 $^{a}$ ± 0.12 |
| | \multicolumn 2019/2020 growth season | | | | | | | | | | | |
| HS$_0$ | 659.3 ± 4.81 | 705.0 ± 11.0 | 559.2 ± 0.48 | 28.7 $^{e-h}$ ± 1.20 | 24.0 $^{kl}$ ± 0.58 | 22.4 $^{lm}$ ± 0.85 | 8.0 $^{f}$ ± 0.46 | 8.0 $^{f}$ ± 0.46 | 11.2 $^{d}$ ± 0.70 | 4.03 $^{fg}$ ± 0.05 | 3.70 $^{gh}$ ± 0.36 | 4.75 $^{ef}$ ± 0.07 |
| HS$_1$ | 621.3 ± 1.91 | 735.1 ± 1.83 | 596.2 ± 31.3 | 29.9 $^{d-g}$ ± 0.26 | 25.8 $^{i-k}$ ± 2.70 | 33.6 $^{bc}$ ± 0.79 | 10.6 $^{de}$ ± 0.21 | 11.8 $^{d}$ ± 0.35 | 15.5 $^{b}$ ± 0.59 | 4.14 $^{fg}$ ± 0.03 | 2.52 $^{ij}$ ± 0.44 | 5.09 $^{de}$ ± 0.07 |
| HS$_2$ | 728.3 ± 7.22 | 551.8 ± 15.7 | 822.0 ± 29.1 | 31.3 $^{c-e}$ ± 0.52 | 28.8 $^{e-h}$ ± 0.94 | 37.4 $^{a}$ ± 0.49 | 8.0 $^{f}$ ± 0.18 | 16.0 $^{b}$ ± 0.35 | 19.8 $^{a}$ ± 0.25 | 3.94 $^{fg}$ ± 0.06 | 5.23 $^{de}$ ± 0.23 | 9.40 $^{a}$ ± 0.13 |
| HS$_3$ | 660.7 ± 13.2 | 683.9 ± 30.0 | 590.5 ± 31.0 | 34.0 $^{b}$ ± 1.13 | 18.5 $^{n}$ ± 0.40 | 27.4 $^{g-i}$ ± 1.25 | 8.6 $^{f}$ ± 0.74 | 13.8 $^{c}$ ± 0.92 | 8.3 $^{f}$ ± 0.34 | 3.12 $^{hi}$ ± 0.47 | 3.96 $^{fg}$ ± 0.24 | 5.71 $^{cd}$ ± 0.08 |
| HS$_4$ | 535.8 ± 0.53 | 640.6 ± 31.7 | 603.7 ± 31.7 | 26.9 $^{h-j}$ ± 0.61 | 31.6 $^{b-d}$ ± 0.77 | 28.5 $^{f-i}$ ± 0.42 | 9.3 $^{ef}$ ± 0.35 | 13.4 $^{c}$ ± 0.79 | 6.2 $^{g}$ ± 0.39 | 4.22 $^{fg}$ ± 0.43 | 6.21 $^{c}$ ± 0.10 | 7.96 $^{b}$ ± 0.11 |
| HS$_5$ | 538.4 ± 5.66 | 485.2 ± 41.1 | 782.8 ± 41.1 | 24.5 $^{j-l}$ ± 0.57 | 27.9 $^{g-i}$ ± 0.39 | 33.1 $^{bc}$ ± 0.70 | 8.5 $^{f}$ ± 0.46 | 10.7 $^{de}$ ± 0.18 | 15.5 $^{b}$ ± 0.59 | 5.66 $^{cd}$ ± 0.57 | 4.65 $^{ef}$ ± 0.18 | 4.80 $^{ef}$ ± 0.07 |
| HS$_6$ | 441.0 ± 5.56 | 628.4 ± 4.62 | 70.36 ± 36.8 | 21.2 $^{m}$ ± 0.04 | 28.3 $^{g-i}$ ± 0.38 | 31.2 $^{c-f}$ ± 0.69 | 10.6 $^{de}$ ± 0.18 | 16.5 $^{b}$ ± 0.28 | 15.2 $^{b}$ ± 0.58 | 2.16 $^{j}$ ± 0.03 | 6.08 $^{c}$ ± 0.06 | 7.52 $^{b}$ ± 0.11 |

Means in the same column denoted by a different letter indicate significant difference between treatments at ≤0.05. HS$_1$ = Humic substances applied as foliar spray at 1.0 g L$^{-1}$, HS$_2$ = Humic substances applied as foliar spray at 2.0 g L$^{-1}$, HS$_3$ = Humic substances applied as foliar spray at 4.0 g L$^{-1}$, HS$_4$ = Humic substances applied as soil application at 5.04 kg ha$^{-1}$, HS$_5$ = Humic substances applied as soil application at 7.56 kg ha$^{-1}$, HS$_6$ = Humic substances applied as soil application at 10.08 kg ha$^{-1}$, V$_1$ = Seds1 cv, V$_2$ = Misr2 cv, V$_3$ = Giza168 cv, Fe = iron content, Mn = Manganese content, Zn = Zinc content and Cu = cooper content.

### 3.5. Pearson's Correlations and Regression Analysis

Pearson's correlation coefficients between total grain yield (TGY) and biological yield (BY) and other important attributes such as grain weight per spike (GWS), grain number per spike (GNS), leaf nitrogen (N), phosphorus (P), potassium (K), iron (Fe), manganese (Mn), zinc (Zn) and copper (Cu) contents are presented in Table 10. High significant ($p \leq 0.01$) positive correlations ($r = 0.672$ **, $0.804$ **, $0.588$ ** and $0.433$ **) were obtained between TGY and each of BY, GWS, GNS and Mn, respectively. In addition, significant ($p \leq 0.05$) positive correlations ($r = 0.307$ *) were found for TGY and Zn. Furthermore, TGY had highly significant negative correlations ($r = -0.417$ **) with K and significant negative correlations ($r = -0.208$ *) with N in the 2018/2019 season. However, TGY had highly significant positive correlations ($r = 0.499$ **) with BY and a significant positive correlation ($r = 0.262$ *) with Mn in the 2019/2020 season.

**Table 10.** A matrix of linear correlation coefficients and their significance levels between total grain yield (TGY, t ha$^{-1}$) and other important attributes and nutrients of some wheat varieties grown in clay loam soil in 2018/2019 and 2019/2020 seasons.

| | C | 1 | 2 | 3 | 4 | 5 | 6 | 7 | 8 | 9 | 10 | 11 |
|---|---|---|---|---|---|---|---|---|---|---|---|---|
| | | | | | | **2018/2019 season** | | | | | | |
| 1 | TGY | 1 | 0.672 ** | 0.804 ** | 0.588 ** | −0.208 * | 0.056 | −0.417 ** | 0.101 | 0.433 ** | 0.307 * | 0.228 |
| 2 | BY | | 1 | 0.220 | 0.316 * | −0.455 ** | −0.131 | −0.533 ** | 0.160 | 0.510 ** | 0.166 | 0.296 ** |
| 3 | GWS | | | 1 | 0.569 ** | −0.024 | 0.100 | −0.176 | 0.048 | 0.276 ** | 0.303 * | 0.081 |
| 4 | GNS | | | | 1 | −0.488 ** | 0.164 | −0.494 ** | 0.325 ** | 0.211 | 0.033 | 0.203 |
| 5 | N | | | | | 1 | 0.229 | 0.630 ** | −0.433 ** | −0.187 | −0.061 | −0.454 ** |
| 6 | P | | | | | | 1 | 0.097 | 0.082 | −0.259 * | 0.178 | −0.277 ** |
| 7 | K | | | | | | | 1 | −0.424 ** | −0.325 ** | 0.113 | −0.359 ** |
| 8 | Fe | | | | | | | | 1 | 0.064 | 0.175 | 0.328 ** |
| 9 | Mn | | | | | | | | | 1 | 0.242 | 0.099 |
| 10 | Zn | | | | | | | | | | 1 | 0.398 ** |
| 11 | Cu | | | | | | | | | | | 1 |
| | | | | | | **2019/2020 season** | | | | | | |
| 1 | TGY | 1 | | | | | | | | | | |
| 2 | BY | 0.499 ** | 1 | | | | | | | | | |
| 3 | GWS | 0.216 | 0.053 | 1 | | | | | | | | |
| 4 | GNS | 0.128 | −0.052 | 0.623 ** | 1 | | | | | | | |
| 5 | N | 0.024 | 0.064 | −0.032 | −0.056 | 1 | | | | | | |
| 6 | P | −0.169 | 0.026 | −0.560 ** | −0.188 | 0.076 | 1 | | | | | |
| 7 | K | −0.136 | −0.163 | 0.021 | 0.008 | 0.012 | −0.024 | 1 | | | | |
| 8 | Fe | 0.165 | 0.020 | 0.078 | −0.215 | 0.077 | −0.170 | −0.018 | 1 | | | |
| 9 | Mn | 0.262 * | 0.285 * | 0.115 | −0.202 | 0.309 * | −0.307 * | −0.044 | 0.435 ** | 1 | | |
| 10 | Zn | 0.223 | 0.309 * | −0.006 | −0.112 | 0.148 | −0.036 | −0.037 | 0.346 ** | 0.346 ** | 1 | |
| 11 | Cu | 0.215 | 0.341 * | 0.036 | −0.239 | −0.167 | −0.008 | 0.029 | 0.248 | 0.449 ** | 0.397 ** | 1 |

* $p \leq 0.05$, ** $p \leq 0.01$ and ns-non significant. TGY = Total grain yield, BY = Biological yield, GWS = Grain weight spike$^{-1}$, GNS = Grain number spike$^{-1}$, N = nitrogen, P = Phosphorus, K = potassium, Fe = iron, Mn = manganese, Zn = zinc and Cu = copper.

Stepwise regression analysis in Table 11. indicated that the variations in TGY among wheat varieties are explained by variations in the growth and yield attributes (i.e., GWS, PH and SPAD reading) in the 2018/2019 seasons, and by variations in leaf Mn contents in the 2019/2020 season.

**Table 11.** Proportional contribution in predicting total grain yield (TGY) using stepwise multiple linear regression for some wheat varieties plants applied with different humic substances levels in 2018/2019 and 2019/2020 seasons.

| Year | r | $R^2$ | Adjusted $R^2$ | SEE | Sig. | Fitted Equation |
|---|---|---|---|---|---|---|
| 2018–2019 | 0.881 | 0.777 | 0.765 | 0.627 | *** | TGY = −4.903 + 1.97 GWS + 0.19PH + 0.073SPAD |
| 2020–2021 | 0.260 | 0.069 | 0.054 | 1.091 | *** | TGY = 5.495 + 0.063 Mn |

*** Indicates differences at $p \leq 0.001$ probability level. SEE = Standard error of the estimates.

## 4. Discussion

Soil pH is an important indicator of the chemistry and fertility of soil due to its key role in the availability and mobility of nutrients in the soils, thus indicating soil fertility and nutritional status for plants. As shown in Table 4, the tested soil suffered from increasing soil pH (7.66 and 7.96), thus decreasing the availability of macro- and micronutrients, despite their presence in the soil. This study was conducted to evaluate the performance of some wheat varieties under high soil pH conditions when different humic substances (HS) were applied, either as a foliar spray or as a soil application. In recent years, the application of HSs has been increased by several types of growers, both small growers and agricultural business, as well as different production systems (conventional and organic cropping) due to their important role in improving soil conditions and achieving equilibrium among nutrients in the soil, affecting soil productivity and crop production. The obtained data (Table 5) indicated that most vegetative growth and physiological parameters were influenced by HS treatments, irrespective of the application method; these results are in accordance with the findings of [25]. This study concluded that HA had a positive influence on growth parameters due to its beneficial role, directly or indirectly, on plant growth and development. This was a result of its effects on the cell membrane and oxygen uptake, which led to enhanced transport of ions, increased protein and carbohydrate synthesis, plant hormone-like activity, enhanced photosynthesis, modified enzyme activities, increased nutrient and water uptake, and reduced toxic effects of some high element levels in the soil [26]. These results may be further explained by the ability of HS to promote the growth of plant roots and root hairs, thus enhancing the ability of plants to take up water and nutrients from the soil, so that plants can develop better. Furthermore, by reducing the evaporation of water by controlling the opening and closing of the leaf stomata, either plants or the soil can maintain more water and thus increase some vegetative growth parameters [27]. In other words, pronounced increases in growth parameters such as plant height and leaf area could be due to an increase in nutrient uptake and a consequent increase in metabolic processes within the plants that produce several organic compounds. These, in turn, result in increases in cell elongation and division and act as catalysts for the activity of several plant hormones and encourage the release of several types of oxinate [28]. Another explanation was suggested by [29,30], which suggested the essential role of HS in the induction of enzymes related to carbon (C) and nitrogen (N) metabolism, and glutamate dehydrogenase, nitrate reductase and glutamine synthesis. Furthermore, HSs are excellent at stimulating microbial activity [31]. For the SPAD readings, the statistical analysis results showed some non-significant effects, as presented in Figure 3; nevertheless, HS caused remarkable increases in SPAD readings. The authors of [32] explained the impact of HA on the SPAD readings as HA being able stimulate biochemical processes inside plants, such as photosynthesis and total chlorophyll content. The results of this study were in accordance with the findings of the authors of [28], who showed that the SPAD reading increases were due to the HS application increasing nutrient and water uptake, consequently increasing the production of compounds necessary for the production of the chloroplasts responsible for chlorophyll synthesis in plants. Our results indicated that the SPAD reading in the first season was higher than in the second season. This could be due to the fact that the temperature in the first season was more

suitable in the second season, as higher temperatures were observed during the second growing season, as shown in Table 1, which negatively affected chlorophyll synthesis.

With regard to the influence of the HS × variety interaction, our results, as presented in Table 5, indicated that the soil application recorded the highest values compared with the foliar spray. These results could be explained by the tested HS containing fulvic acid (FA), which is characterized by a low molecular weight (MW ≤ 3000.0 Da), as shown in Table 3. FA is completely dissolved in soils and directly absorbed by plant roots [33] due to its small MW. Furthermore, application of FA directly affects the metabolic sites and plant cells, and FA plays a role in the release of nutrients for plants [34]. Some researchers have another explanation: HSs also allow plant growth regulators to regulate hormone levels inside plants and boost the production of plant enzymes, and additionally stimulating enzyme catalysis as well as enhancing respiration and photosynthesis [35–37].

The impact of HS, either as a foliar spray or a soil application, on yield and its attributes is outlined in Table 6. The results of this study could be explained by an increase in wheat leaf nitrogen (N) and phosphorus (P) contents, as shown in Table 8. Both nutrients improved the meristematic activities and cell division, which enhanced the wheat growth parameters and photosynthesis processes, and, consequently, the formation and filling of grains [38]. In the same context, some researchers [39,40] demonstrated the desirable impact of N on these attributes, as N is the major nutrient for plant nutrition and causes increases in vegetative growth for plants and forms strong plants with long spikes. Furthermore, N encourages plants to take up other nutrients such as iron (Fe), manganese (Mn), zinc (Zn) and copper (Cu), therefore enhancing plant growth and, consequently, yield attributes. In addition, refs. [41,42] concluded that HSs play a vigorous role in nutrient uptake by plants through direct and indirect influences, increasing the permeability of membranes in the root cells. The results of our investigation could be explained by the key role of HS in freeing up nutrients in the soil. The other beneficial impacts of HS included improved water and nutrient uptake through its effects on the soil's chemical, physical and biological properties and the decrease in soil pH, as it acts as a chelating agent resulting from the presence of several carboxylic groups (R-COOH) in its structure, so that nutrients become available to the root and root hairs of plants. A further alternative explanation could be the main role of HS; however, HS induced plants to produce some plant regulators, i.e., indole acetic acid (IAA), gibberellins and cytokinins, which affected root development. In addition, HS was able to increase the permeability of plant cell membranes by its hormone-like activities that enhanced cell respiration, photosynthesis, protein synthesis and various enzymatic reactions, consequently increasing the uptake of micronutrients including Fe, Mn, Zn and Cu. Our results indicated that high levels of HS are less effective and the application of excessive amounts might result in pollution of the environment. These results were in accordance with the findings of [43,44]. However, HS not only caused pronounced increases in grain yield but also enhanced yield attributes through its influence on the cell membranes and oxygen uptake, thus increasing respiration and photosynthesis as well as root cell elongation [45]. Furthermore, HS alleviated some toxic impacts of high levels of some minerals inside plants [46] and in the soil [47,48]. On the other hand, some researchers mentioned that HSs play a vital role in plant growth through the stimulation of root growth, consequently increasing water uptake and nutrient absorption [12], in addition to enhancing protein synthesis [11,49], which is reflected in the yield and its attributes. It was noted that the results of yield attributes such as GWS, GNS and TGW were not identical to TGY, as the yield attributes increased during the 2018/2019 season, while the TGY was higher during 2019/2020. This is due to the presence of some other parameters that have affected TGY, such as increases in the number of lateral branches and the number of spikes per plant during harvest time. All these factors could have caused an increase in TGY in the first season compared to the second season. Thus, GWS, GNS and TGW are not the only limiting parameters for TGY. As shown in Figure 14, the differences of biological yield between control treatment in both seasons are very slight and were only highly significant in the second season. The decrease in the biological yield values, as presented in Figures 17 and 18,

is due to the inability of the studied wheat varieties to adapt to weather conditions in the 2019/2020 season. On the contrary, our results regarding leaf potassium (K) contents are not in accordance with the findings of [49–51]. However, untreated plants (HS$_0$) had a decrease in leaf K contents due to the antagonism with leaf N and K contents.

Data related to the leaf macro-and micronutrient contents revealed that HS treatments increased leaf nutrient contents. These results are in agreement with [50,52]. They suggested that the availability of P is a consequence of HS protecting the P from precipitation and enhancing its uptake by complexing with Fe. On the contrary, leaf N contents in the first season and K contents in both seasons decreased. The authors of [53,54] noted that the effect of HS might be partially due to different soils. In Figure 21, leaf potassium content in the second season was higher than in the first season, which might be because the soil's available potassium content in the second season (376 mg kg$^{-1}$) was higher than in the first season (352 mg kg$^{-1}$), as shown in Table 4. In Figure 19, the leaf nitrogen content in the second season was higher than in the first season, a result that could be explained by the antagonism between nitrogen and potassium; however, excessive amounts of nitrogen reduce the uptake of potassium (Ranade-Malvi, 2011) [55]. The results of our experiment indicated that leaf Fe, Mn, Zn and Cu contents increased with HS application as a result of HS ability to complex metal ions and form complexes with micronutrients [56]. Previous studies noted that the increase in leaf micronutrient content could be due to HS acting as a natural chelate for cationic micronutrients; however, the addition of HS may enhance micronutrient absorption by chelating free forms of these micronutrients [57]. Moreover, ref. [58] suggested that cationic micronutrients could be related to hydrogen bonds inside the structure of HS. Furthermore, ref. [59] suggested that typically, organic acids, such as citric, oxalic, tartaric and $\alpha$-glutaric acids, in root exudates can extract Fe, Mn, Zn and Cu from humic acid complexes. Arguably, Fe and Zn could insolubly precipitate with P. On the contrary, the leaf nutrient contents decreased under high levels of HS as a foliar spray in [60], which suggested that these decreases could be because excessive doses of HS might cause over-complexations with these nutrients. We suggest another mechanism: a high level of HS decreases the conversion of Fe$^{+++}$ to Fe$^{++}$ in the soil.

Concerning the impact of varieties and their capacity for nutrient uptake, most researchers agreed that the variation among wheat varieties could be due to genetic variation and the genetics of the mode of use of metabolic products, as well as the structural ability of some varieties to absorb these nutrients rather than others [61] or the genetic differences in the pedigree of the studied varieties [62–65]. These results are partially in line with those reported by [66–68]. Similar results regarding varietal differences were reported by [69–71].

## 5. Conclusions

Technically, humic substances (HSs) are not fertilizers, although most farmers do consider them so, and their applications are environmentally friendly; furthermore it is considered one of the most important production factors in sustainable agriculture practice. It can be concluded from this study that the application of HSs, irrespective of the method of application, improved the nutritional status of the studied wheat varieties, thus affecting most yield attributes. This study proved that the plants treated with 1.0 (HS$_1$) and 2.0 g L$^{-1}$ (HS$_2$) as a foliar spray and 7.56 (HS$_5$) and 10.08 kg ha$^{-1}$ (HS$_6$) as a soil application were the superior treatments for the most leaf nutrient contents. However, the soil application treatments worked better than foliar spray in terms of growth parameters. In addition, the HS$_1$, HS$_4$ and HS$_5$ treatments produced the maximum values for yield and its attributes. Moreover, Giza168 (V$_3$) showed the largest response to HS, followed by Seds1 (V$_1$) and Misr2 (V$_2$).

**Author Contributions:** Conceptualization, A.A.M.A. and A.B.A.E.-T.; data curation, A.A.M.A. and A.A.A.S.; formal analysis, A.A.M.A., A.B.A.E.-T. and A.A.M.O.; investigation, A.A.M.A., A.B.A.E.-T., A.A.A.S. and A.A.M.O.; methodology, A.A.M.A., A.B.A.E.-T., A.A.A.S. and A.A.M.O.; resources, A.A.M.A., A.B.A.E.-T. and A.A.A.S.; software, A.A.M.A., A.B.A.E.-T., A.A.A.S. and A.A.M.O.; writing-original draft, A.A.M.A., A.B.A.E.-T. and A.A.A.S.; writing-review and editing, A.A.M.A., A.B.A.E.-T., A.A.A.S. and A.A.M.O. All authors have read and agreed to the published version of the manuscript.

**Funding:** This research received no external funding.

**Institutional Review Board Statement:** Not appreciable.

**Informed Consent Statement:** Not appreciable.

**Data Availability Statement:** The data presented in this study are available upon request from the corresponding author.

**Conflicts of Interest:** The authors declare no conflict of interest.

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
