# Peer review of "Nutrient Contents and Productivity of Triticum aestivum Plants Grown in Clay Loam Soil Depending on Humic Substances and Varieties and Their Interactions"

_agronomy, doi:10.3390/agronomy12030705_

Round 1
Reviewer 1 Report
1 Line 23: As three varieties were selected, more results among different varieties should be given.
2 Line 50-51: The relative reference should be added.
3 Line 67-68: “HA is known to be among the most biochemically active materials found in soil and is considered to be the most abundant naturally”?
4 Line 69: “correct” used in there is not suitable. Use “change” instead of “correct” is much better.
5 Line 86: before this paragraph, more information about the use of HS in agriculture and what is very important but not have been studied before should be given.
- the authors should give the reason why you selected these three varieties?
- Line 96: “seven levels of humic substances (HS).” Six?
- Line 107-108: “1.0, 2.0 or 4.0 gL-1 as a foliar spray; 5.04, 7.56 and 10.08 kg ha-1 as a soil application.” the contents selected in this study based on what standard.
- in the 2.2 part, the line spacing and sowing rate are missing.
- Line 114-116: “The plants were fertilized with different fertilizers (N, P and K) according to the recommendations of the Egyptian 115 Ministry of Agriculture.” The specific amount of N, P and K fertilizers used in this study should be added.
- How did the soil application of HS was performed?
- Line 146-149: how did the authors select samples to measure the plant height, SPAD or leaf area?
- Line 159-162: this part should be rewritten. For each index, how many replicates in each plot should be clear.
- Line 190 and the title of this manuscript: did the soil type “clay loam soil” was a very different4 soil with other studies about HS? If so, more information should be given in the introduction part, if not, I recommend deleted “clay loam”.
- for figure 1, 3, the plant height, SPAD in the two years were not changed too much in the control treatment, but declined dramatically in the second year. The potential reasons should be analyzed.
- There are too many figures or tables contain same or similar data, I recommend combined or delete the repetitive ones.
- Figure : the spike length, grain weight spike-1 , grain number spike-1 in the first year were higher than in the second year, but the grain weight in the 2018/2019 season was dramatically lower than in the 2019/2020, the reasons should be analyzed.
- Figure 14, 17, 18: the biological yield in the two season did not changed too much in the control treatment, but it largely declined in the 2019/2020 season than in the 2018/2019 season. Please give the possible reasons.
- Figure 19: The application of HS reduced the leaf nitrogen contents in the first season, but increased it the second season. Why this happened? The same as leaf potassium contents in figure
- Line 683-686: why not all use the two seasons’ data together? but use three indexes in the 2018/2019 season, but another one in the 2019/2020 season? And these four figures should be combined with table 10.
- the discussion part should be rewritten with good logic and based on the main important results and possible reasons to explain the unusual phenomenon.
Author Response
Agronomy - MDPI
Manuscript ID: Agronomy- ID -1569684
Manuscript Title: "Nutrient Contents and Productivity of Triticum aestivum Plants Grown in Clay Loam Soil Depending on Humic Substances and Varieties and their Interactions"
=====================================================================
Thank you for your efforts and I would like also to thank very much the reviewers for their valuable comments. We have corrected the manuscript based on the comments of reviewers, and the corrections made in the text in red color, and are outlined step by step as follows:
Response to the comments of Reviewer 1:
Moderate English changes required.
The manuscript entitled "Nutrient Contents and Productivity of Triticum aestivum Plants Grown in Clay Loam Soil Depending on Humic Substances and Varieties and their Interactions" is a well-written manuscript by the authors.
1- Line 23: As three varieties were selected, more results among different varieties should be given
Re: V3 (Var168) is one of the high-yield varieties that withstand high temperatures and irrigation water deficit compared to other varieties. V1 (Seds1) is characterized by high yield in addition to its high tolerance to yellow rust disease. For V2 (Misr2) is characterized by its high production and its tolerance to salinity and water deficit. Based on these seasons, these three varieties are the most widespread and cultivated in that region (lines 102-108)
2- Line 50-51: The relative reference should be added.
Re: Done [Ghanem et al., 2021] in line 52
3- Line 67-68: HA is known to be among the most biochemically active materials found in soil and is considered to be the most abundant naturally?
Re: According to Webe, 2020. I added the reference.
4- Line 69: "correct" used in there is not suitable. Use "change" instead of "correct" is much better
Re: I used “change” instead of “correct” in line 71
5- Line 86: before this paragraph, more information about the use of HS in agriculture and what is very important but not have been studied before should be given.
Re: The research has a set of objectives, including trying to reduce the amount of chemical fertilizers used and replacing them with humic substances, in addition to studying the optimal rate and the best applying to add HS for wheat crop under Aswan conditions as shown in lines 89-93.
6- the authors should give the reason why you selected these three varieties?
Re: we mentioned the reason why we selected these varieties in lines 101-103.
7- Line 96: “seven levels of humic substances (HS).” Six?
Re: I changed from “seven” to “six’ in line 98.
8- Line 107-108: “1.0, 2.0 or 4.0 gL-1 as a foliar spray; 5.04, 7.56 and 10.08 kg ha-1 as a soil application.” the contents selected in this study based on what standard.
Re: In foliar spray; the doses were determined based on the total amount of water required per hectare. While, in soil application according to the minimum recommended level for cereal crops.
9- in the 2.2 part, the line spacing and sowing rate are missing.
Re: I mentioned the line spacing and sowing rate in lines 117-119.
10- Line 114-116: “The plants were fertilized with different fertilizers (N, P and K) according to the recommendations of the Egyptian 115 Ministry of Agriculture." The specific amount of N, P and K fertilizers used in this study should be added.
Re: The specific amount of N, P and K fertilizers used should be added in lines 120-122
11- How did the soil application of HS was performed?
Re: the humic materials were dissolved in water and applied to the plants
12- Line 146-149: how did the authors select samples to measure the plant height, SPAD or leaf area?
Re: the readings of plant height, SPAD and leaf area were obtained from 20 plants from each plot as shown in lines 153and 154.
13- Line 159-162: this part should be rewritten. For each index, how many replicates in each plot should be clear
Re: 3 replicates were mentioned in line 125. In addition, I wrote it again in line 160.
14- Line 190 and the title of this manuscript: did the soil type “clay loam soil” was a very different4 soil with other studies about HS? If so, more information should be given in the introduction part, if not, I recommend deleted “clay loam”
Re: There are no differences in soil type.
15- For figure 1, 3, the plant height, SPAD in the two years were not changed too much in the control treatment, but declined dramatically in the second year. The potential reasons should be analyzed.
Re: As shown in table 1, related to the climatic conditions in the region of study. It is noted that the average temperatures °C (day and night) are higher in the second season than in the first season. As it is known that wheat plants grow in cooler temperature preferentially, however the optimum temperature for growth range 15-25 °C (Mueller et al., 2015). Â
16- There are too many figures or tables contain same or similar data, I recommend combined or delete the repetitive ones
Re: I deeply apologize as i am sure that the manuscript is long and i’ll take that in the next time, but all the studied parameters in this manuscript are very important, especially since that the manuscript in plant nutrition scope, and therefore it is important to study all the factors that affect the yield. It is difficult to combine the data in both growth seasons due to the differences in temperature during both growth seasons.
17- Figures : the spike length, grain weight spike-1, grain number spike-1 in the first year were higher than in the second year, but the grain weight in the 2018/2019 season was dramatically lower than in the 2019/2020, the reasons should be analyzed.
Re: Considering the macro and micro nutrient leaf contents of wheat plants, The results indicated that the contents of the leaves of these elements are in line with the values ​​of the total yield, as the height of the leaf content of these elements in the first season was higher than the second season and thus the yield increased in the second season over the first season (lines 758-766)
18- Figure 14, 17, 18: the biological yield in the two season did not changed too much in the control treatment, but it largely declined in the 2019/2020 season than in the 2018/2019 season. Please give the possible reasons
Re: This difference could be due to the genetically variations among the studied varieties and their interactions with weather conditions, rates and methods of applying of humic acids (lines 766-770)
19- Figure 19: The application of HS reduced the leaf nitrogen contents in the first season, but increased it the second season. Why this happened? The same as leaf potassium contents in figure.
Re: in figure leaf potassium content in the second season was the higher than the first season which may be due to soil available potassium content in the second season (376 mgkg-1) was higher than the first season (352 mgkg-1) as shown in table 4. While, in figure 19, the leaf nitrogen contents in the second season was higher than the first season, these results could be explained due to the antagonism between nitrogen and potassium, however the excessive amounts of nitrogen reduce the uptake of potassium (Ranade-Malvi, 2011) (lines 782-789).
20- Line 683-686: why not all use the two seasons’ data together? but use three indexes in the 2018/2019 season, but another one in the 2019/2020 season? and these four figures should be combined with table 10.
Re: I replaced all the stepwise figures (No 33-36) with table 11 to.
21- the discussion part should be rewritten with good logic and based on the main important results and possible reasons to explain the unusual phenomenon.
Re: I rewrote the discussion again based on your comments and reply all inquiries as shown in red color
Many thanks to Reviewer 1 for his valuable comments

Reviewer 2 Report
1 Line 42-44: This sentence seems not connected well with the above one.
2 Line 46-50: This sentence should be rewritten, not very clear, or in other word, looks confused.
- Line 72: changed “however” to “moreover”.
- The innovation of this study was not very clear in the information part.
- Line 95-96: what the difference among the three wheat varieties?
- Line 111: “The main experimental plots were arranged by the three wheat varieties” as the variety was set as the main experiment plot. The results among different varieties should be described in the abstract.
- Line 113-114: the linewidth and seeding rate should also be given.
- Line 129,130…: “as described by [17]”, given the first author’s name is much better than the reference number.
- Line 148-149: which leaf was selected to measure the SPAD or leaf area?
- Line 190 and other figure captions: why the soil type was emphasized?
- Line 190: “bars with a …” changed to “bars in the same year with a …”
- Line 191-195: “HS1 = HS 191 applied as foliar spray at 1.0 gL-1, HS2 = HS applied as foliar spray at 2.0 gL-1, HS3 = HS 192 applied as foliar spray at 4.0 gL-1, HS4 = HS applied as soil application at 5.04 kg ha-1, HS5 193 = HS applied as soil application at 7.56 kg ha-1, HS6 = HS applied as soil application at 194 10.08 kg ha-1.” Was changed to “HS1 , HS2, HS3 represent HS applied as foliar spray at 1.0, 2.0, 4.0 gL-1, respectively, HS4, HS5, HS6 represent HS applied as soil application at 5.04, 7.56, 10.08 kg ha-1.”. the same as other figures.
- Table 5 contains the same data as in figure 1-6, and the interaction between humic substances and varieties is difficult to read, and the meaning of different letters in this table should be cleared.
- why the grain weight increased in the second season when the other indexes about spike or grain decreased?
- Figure 8: changed “grain weight spike-1” to “grain weight per spike”. The same as Figure 9.
- Figure 20: for the2 H0 treatment, the leaf phosphorus contents in the 2019/2020 sesason was largely higher than in the 2018/2019 season, however, after application of HS, it declined dramatically in several treatments. Please give the possible reasons.
- There are too many tables and figures. Many of them should be merged or set as the supplemental materials.
- the discussion was weak and should be rewritten. The main results and the innovation are not clear.
Author Response
Agronomy - MDPI
Manuscript ID: Agronomy-1569684
Manuscript Title: "Nutrient Contents and Productivity of Triticum aestivum Plants Grown in Clay Loam Soil Depending on Humic Substances and Varieties and their Interactions"
=====================================================================
Thank you for your efforts and I would like also to thank very much the reviewers for their valuable comments. We have corrected the manuscript based on the comments of reviewers, and the corrections made in the text in red color, and are outlined step by step as follows:
Response to the comments of Reviewer 2:
- Line 42-44: This sentence seems not connected well with the above one
Re: this sentence transfer in the next paragraph (lines 50 and 51)
- Line 46-50: This sentence should be rewritten, not very clear, or in other word, looks confused.
Re: I rewrote the sentence again, however I added the substitution of some organic fertilizers such as humic substances (HS), whether partially or completely
- Line 72: changed "however" to "moreover".
Re: I changed "however" to "moreover"
- The innovation of this study was not very clear in the information part.
Re: Despite the numerous studies that have been conducted on the impact of humic acid, very few studies have not addressed the comparison between the different application methods (foliar and soil application) as elucidated in the last paragraph of introduction section, this is the main objective of the study Lines 67-68) (line 89-93).
5. Line 95-96: what the difference among the three wheat varieties?
Re: V3 (Var168) is one of the high-yield varieties that withstand high temperatures and irrigation water deficit compared to other varieties. V1 (Seds1) is characterized by high yield in addition to its high tolerance to yellow rust disease. While V2 (Misr2) is characterized by its high production and its tolerance to salinity and water deficit. Based on these seasons, these three varieties are the most widespread and cultivated in that region
- Line 111: "The main experimental plots were arranged by the three wheat varieties" as the variety was set as the main experiment plot. The results among different varieties should be described in the abstract.
Re: I added the statistical analysis design in the abstract (lines 18-20).
- Line 113-114: the line width and seeding rate should also be given.
Re: I mentioned that plot area =10.5 m2 (3 m width) x (3.5 m length) as shown in line 118.
- Line 129,130..: "as described by [17]", given the first author’s name is much better than the reference number
Re: Done I wrote, "Page et al." instead of “17†in lines 139 and 146.
- Line 148-149: which leaf was selected to measure the SPAD or leaf area?
Re: I mentioned that the 4 and 5th leaves were selected in line 157
- Line 190 and other figure captions: why the soil type was emphasized?
Re: As mentioned in the title of manuscript. The soil texture is very important, however the influence humic substances varies according to the soil type. In addition part of our research plan including the influence of humic substances in different texture soils.
- Line 190: “bars with a …†changed to “bars in the same year with a …â€
Re: done in all figure captions
- Line 191-195: “HS1 = HS 191 applied as foliar spray at 1.0 gL-1, HS2 = HS applied as foliar spray at 2.0 gL-1, HS3 = HS 192 applied as foliar spray at 4.0 gL-1, HS4 = HS applied as soil application at 5.04 kg ha-1, HS5 193 = HS applied as soil application at 7.56 kg ha-1, HS6 = HS applied as soil application at 194 10.08 kg ha-1.†Was changed to “HS1 , HS2, HS3 represent HS applied as foliar spray at 1.0, 2.0, 4.0 gL-1, respectively, HS4, HS5, HS6 represent HS applied as soil application at 5.04, 7.56, 10.08 kg ha-1.â€. the same as other figures.
Re: Changed in all figures
- Table 5 contains the same data as in figure 1-6, and the interaction between humic substances and varieties is difficult to read, and the meaning of different letters in this table should be cleared.
Re: I redesign the table 5 and reduce the standard error (SE) to one decimal place to make it easier for the readers
- why the grain weight increased in the second season when the other indexes about spike or grain decreased?
Re: Considering the macro and micro nutrient leaf contents of wheat plants, The results indicated that the contents of the leaves of these elements are in line with the values ​​of the total yield, as the height of the leaf content of these elements in the first season was higher than the second season and thus the yield increased in the second season over the first season (lines 758-766)
- Figure 8: changed “grain weight spike-1†to “grain weight per spikeâ€. The same as Figure 9.
Re: done in figures 8 and 9
- Figure 20: for the H0 treatment, the leaf phosphorus contents in the 2019/2020 season was largely higher than in the 2018/2019 season, however, after application of HS, it declined dramatically in several treatments. Please give the possible reasons.
Re: due to the antagonism between Fe and Zn with P as mentioned in line 806
- There are too many tables and figures. Many of them should be merged or set as the supplemental materials.
Re: I know very well that our manuscript is very long and this is what I mentioned to reviewers 3 and 4, and its reasons, but I tried as much as possible to reduce the volume of the manuscript by replacing table 11 instead of four figures (32-35) and I have no objection to merging the results obtained, but that requires a change in the discussion, in the end the final decision is yours.
- The discussion was weak and should be rewritten. The main results and the innovation are not clear.
Re: Re: I rewrote the discussion again based on your comments and reply all inquiries as shown in red color
Many thanks to Reviewer#2 for his valuable comments
Reviewer 3 Report
The topic of the manuscript matches the scope of the journal very well. The introduction provides a good, generalized background. Establishes the originality of the research aims by demonstrating the need for investigations in the topic area. The objective is clearly defined. This is followed by a detailed description of experimental methods. Results are referenced with obtained results summarized in figures and tables. Clarity and data presentation quality would be improved. For example: Table 17 – missing units in the table. The tables/figures nicely summarize results analyzed by the authors, but why not support this data with data obtained for the growing seasons 2018/19-2019/20?
The discussion explores the significance of the results in the context of relevant published papers and highlights the key original findings based on data over a period of two growing seasons. The conclusions of the study are presented in a short section highlighting the key results and their significance. The literature cited in the work is relevant to the study. The paper refers to 71 prior art publications, all of which are reasonably well referenced in the text.
I have the following remarks, which can be used by the authors to improve the work:
1) In Table 1 the authors list the weather conditions which accompanied the experiment. Can this data be commented in the manuscript? This is currently a missing part of the discussion.
2) The variable in the experiment is the plant varieties. Can the authors provide more detailed characteristics of these varieties? This would increase the clarity for the benefit of the readers.
3) The results are showing two growing seasons separately (2018/19 and 2019/20). What is missing here is the average from the data on these two seasons separately.
4) How was the grain yield calculated?
5) Did the authors calculate the plant density or number of spikes per 1m2 during harvest and what in the authors’ opinion does the term yield components mean?
Author Response
Agronomy - MDPI
Manuscript ID: Agronomy-1569684
Manuscript Title: "Nutrient Contents and Productivity of Triticum aestivum Plants Grown in Clay Loam Soil Depending on Humic Substances and Varieties and their Interactions"
=====================================================================
Dear Ms. Svjetlana Novković
Special Issue editor, MDPI Novi Sad
        Thank you for your efforts and I would like also to thank very much the reviewers for their valuable comments. We have corrected the manuscript based on the comments of reviewers, and the corrections made in the text in red color, and are outlined step by step as follows:
Response to the comments of Reviewer 3:
Comments and Suggestions for Authors
The topic of the manuscript matches the scope of the journal very well. The introduction provides a good, generalized background. Establishes the originality of the research aims by demonstrating the need for investigations in the topic area. The objective is clearly defined. This is followed by a detailed description of experimental methods. Results are referenced with obtained results summarized in figures and tables. Clarity and data presentation quality would be improved. For example: Table 17–missing units in the table. The tables/figures nicely summarize results analyzed by the authors, but why not support this data with data obtained for the growing seasons 2018/19-2019/20?
In this paragraph what do you mean Table 17?
The discussion explores the significance of the results in the context of relevant published papers and highlights the key original findings based on data over a period of two growing seasons. The conclusions of the study are presented in a short section highlighting the key results and their significance. The literature cited in the work is relevant to the study. The paper refers to 71 prior art publications, all of which are reasonably well referenced in the text. I have the following remarks, which can be used by the authors to improve the work:
- In Table 1 the authors list the weather conditions which accompanied the experiment. Can this data be commented in the manuscript? This is currently a missing part of the discussion
Re: This study was carried out in Aswan district, this province is characterized by high temperature and low rainfall. On the contrast, optimal growth of wheat plants needs low temperatures. Therefore, this table has been set that may explain some of the results in the study. Furthermore the values in table 1 were used to explain some results as mentioned in discussion section. Â
- The variable in the experiment is the plant varieties. Can the authors provide more detailed characteristics of these varieties? This would increase the clarity for the benefit of the readers.
Re: (Var1) Seds1 is characterized by high yield in addition to its high tolerance to yellow rust disease. For (Var2) Misr2 is characterized by its high production and its tolerance to salinity and water deficit. (Var3) Var168 is one of the high-yield varieties that withstand high temperatures and irrigation water deficit compared to other varieties. Based on these seasons, these three varieties are the most widespread and cultivated in that region  Â
- The results are showing two growing seasons separately (2018/19 and 2019/20). What is missing here is the average from the data on these two seasons separately
Re: I miss understanding this comment, so I sent mail to the editor to inquire about this comment exactly. And I am ready to response as soon as the reply arrives.
- How was the grain yield calculated?
Re: It was calculated from the yield of 30 harvested plants and then calculated as a percentage of the total number of plants per hectare (lines 180 and 181)
- Did the authors calculate the plant density or number of spikes per 1m2 during harvest and what in the authors’ opinion does the term yield components mean?
Re: plant density was calculated as shown in the materials and methods section (line 119 and 120) and number of spikes per 1 m2 was also calculated. Yield components are meaning all parameters that affect the yield such as, grain weight, grain number, 1000-grain weight and etc…Â
Many thanks to Reviewer 4 for his valuable comments
Ahmed A.M. Awad (Corresponding author)

Reviewer 4 Report
The manuscript is well done, the experiments were conducted correctly and the authors have a lot of results. Unfortunately, that is a major problem now, because the results need to be summarized in the Figures and Tables. The paper is too long and makes to difficult to the readers understand the real objective and the interpretation the most important results of the entire work. We suggest strongly to reorganize the results under new Figures and Tables. The Discussion is poor and need to be improved also. Regardind the methodology, it is important to demonstrate the level or classification of the nutrients contents in soil, because as an international paper the methods are not common for all the countries. We encourage the authors the rework the manuscript following these recommendations because we are sure that the exposed research is important but needs to be described shortly and objectively. We are anxious for seeing this work in a new form here again.
Author Response
Agronomy - MDPI
Manuscript ID: Agronomy-1569684
Manuscript Title: "Nutrient Contents and Productivity of Triticum aestivum Plants Grown in Clay Loam Soil Depending on Humic Substances and Varieties and their Interactions"
=====================================================================
Thank you for your efforts and I would like also to thank very much the reviewers for their valuable comments.
Response to the comments of Reviewer 4:
Comments and Suggestions for Authors
The manuscript is well done, the experiments were conducted correctly and the authors have a lot of results. Unfortunately, that is a major problem now, because the results need to be summarized in the Figures and Tables. The paper is too long and makes too difficult to the readers understand the real objective and the interpretation the most important results of the entire work. We suggest strongly to reorganize the results under new Figures and Tables. The Discussion is poor and need to be improved also. Regarding the methodology, it is important to demonstrate the level or classification of the nutrients contents in soil, because as an international paper the methods are not common for all the countries. We encourage the authors the rework the manuscript following these recommendations because we are sure that the exposed research is important but needs to be described shortly and objectively. We are anxious for seeing this work in a new form here again.
Re: First of all, I appreciate your time and effort reviewing our manuscript, and i emphasize on taking your comments into consideration. Accordingly, i would like to point out that this manuscript is a part of big research that includes different and various of agricultural specializations; hence, this manuscript takes less than 20% of what the whole thesis includes. However, I am working right now on summarizing and replying all send comments to be suitable for publication. Finally, your effort is too much appreciated and many thanks for your distinguished comments.
Many thanks to Reviewer 4 for his valuable comment
